# Bile acid synthesis, modulation, and dementia: A metabolomic, transcriptomic, and pharmacoepidemiologic study

Vijay R. Varma[1‡], Youjin Wang[2‡], Yang An[3], Sudhir Varma[4], Murat Bilgel[3], Jimit Doshi[5], Cristina Legido-Quigley[6], João C. Delgado[7], Anup M. Oommen[8], Jackson A. Roberts[1], Dean F. Wong[9], Christos Davatzikos[5], Susan M. Resnick[3], Juan C. Troncoso[10], Olga Pletnikova[10], Richard O'Brien[11], Eelko Hak[12], Brenda N. Baak[12], Ruth Pfeiffer[2], Priyanka Baloni[13], Siamak Mohmoudiandehkordi[14], Kwangsik Nho[15], Rima Kaddurah-Daouk[14], David A. Bennett[16], Shahinaz M. Gadalla[2], Madhav Thambisetty[1]*

1 Clinical and Translational Neuroscience Section, Laboratory of Behavioral Neuroscience, National Institute on Aging (NIA), National Institutes of Health (NIH), Baltimore, Maryland, United States of America, 2 Clinical Genetics Branch, Division of Cancer Epidemiology and Genetics, National Cancer Institute, National Institutes of Health, Bethesda, Maryland, United States of America, 3 Brain Aging and Behavior Section, Laboratory of Behavioral Neuroscience, National Institute on Aging (NIA), National Institutes of Health (NIH), Baltimore, Maryland, United States of America, 4 HiThru Analytics, Laurel, Maryland, United States of America, 5 Section for Biomedical Image Analysis, Department of Radiology, University of Pennsylvania, Philadelphia, Pennsylvania, United States of America, 6 Kings College London, United Kingdom, 7 College of Medicine and Health, University of Exeter, Exeter, United Kingdom, 8 Glycoscience Group, NCBES National Centre for Biomedical Engineering Science, National University of Ireland Galway, Galway, Ireland, 9 Department of Radiology, Johns Hopkins University School of Medicine, Baltimore, Maryland, United States of America, 10 Department of Pathology, Johns Hopkins University School of Medicine, Baltimore, Maryland, United States of America, 11 Department of Neurology, Duke University School of Medicine, Durham, North Carolina, United States of America, 12 Groningen Research Institute of Pharmacy, University of Groningen, Groningen, the Netherlands, 13 Institute for Systems Biology, Seattle, Washington, United States of America, 14 Department of Psychiatry and Behavioral Sciences, Department of Medicine, Duke University Medical Center, Durham, North Carolina, United States of America, 15 Department of Radiology and Imaging Sciences and the Indiana Alzheimer Disease Center, Indiana University School of Medicine, Indianapolis, Indiana, United States of America, 16 Rush Alzheimer's Disease Center, Rush University Medical Center, Chicago, Illinois, United States of America

‡ These authors share first authorship on this work.
* thambisettym@mail.nih.gov

**Data Availability Statement:** Baltimore Longitudinal Study of Aging (BLSA) data are available to researchers and can be requested at https://www.blsa.nih.gov/researchers. Rush

## Abstract

### Background

While Alzheimer disease (AD) and vascular dementia (VaD) may be accelerated by hypercholesterolemia, the mechanisms underlying this association are unclear. We tested whether dysregulation of cholesterol catabolism, through its conversion to primary bile acids (BAs), was associated with dementia pathogenesis.

### Methods and findings

We used a 3-step study design to examine the role of the primary BAs, cholic acid (CA), and chenodeoxycholic acid (CDCA) as well as their principal biosynthetic precursor, 7α-hydroxycholesterol (7α-OHC), in dementia. In Step 1, we tested whether serum markers of

Memory and Aging Project (ROSMAP) data can be requested at https://www.radc.rush.edu; ROSMAP data used in this study are available at: https://www.synapse.org/#!Synapse:syn18485175 under the doi 10.7303/syn18485175 Clinical Practice Research Datalink (CPRD) data are available to researchers and can be requested at https://www.cprd.com/public. Alzheimer's Disease Neuroimaging Network (ADNI) data are available to researchers at http://adni.loni.usc.edu.

**Funding:** This research was supported in part by the intramural program of the National Institute on Aging (NIA) and the National Cancer Institute (NCI). ROSMAP is supported by NIA grants P30AG10161, R01AG15819, R01AG17917, and U01AG61356. The ADMC is supported by National Institute on Aging (NIA): grant R01AG046171, a component of the Accelerated Medicines Partnership for AD (AMP-AD) Target Discovery and Preclinical Validation Project; grant RF1 AG0151550, a component of the M2OVE-AD Consortium (Molecular Mechanisms of the Vascular Etiology of AD–Consortium; and RF1AG057452, R01AG059093, RF1AG058942, U01AG061359, U19AG063744 and FNIH: #DAOU16AMPA. Specific authors, indicated in parentheses, were supported by additional grants: NIA RF1 AG058942 and R01 AG057452 (RKD); NLM R01 LM012535 and NIA R03 AG054936 (KN). MT is grateful for funding support from the Andrew and Lillian A. Posey foundation to the Clinical and Translational Neuroscience Section, Laboratory of Behavioral Neuroscience, NIA. The funders had no role in study design, data collection and analysis, decision to publish, or preparation of the manuscript.

**Competing interests:** I have read the journal's policy and the authors of this manuscript have the following competing interests: DFW has prior contracts with Roche Neuroscience, AVID pharma, Lundbeck.

**Abbreviations:** 7α-OHC, 7α-hydroxycholesterol; AD, Alzheimer disease; ADMC, Alzheimer's Disease Metabolomics Consortium; ADNI, Alzheimer's Disease Neuroimaging Initiative; BA, bile acid; BAS, bile acid sequestrants; BBB, blood–brain barrier; BLSA, Baltimore Longitudinal Study of Aging; BLSA-NI, Baltimore Longitudinal Study of Aging Neuroimaging; CA, cholic acid; CB, cerebellum; CDCA, chenodeoxycholic acid; cDVR, cortical distribution volume ratio; CHRM2, Cholinergic Receptor Muscarinic 2; CHRM3, Cholinergic Receptor Muscarinic 3; CI, confidence interval; CON, control; CPRD, Clinical Practice Research Datalink; CSF, cerebrospinal fluid; DCA, deoxycholic acid; EHR, electronic health record;

cholesterol catabolism were associated with brain amyloid accumulation, white matter lesions (WMLs), and brain atrophy. In Step 2, we tested whether exposure to bile acid sequestrants (BAS) was associated with risk of dementia. In Step 3, we examined plausible mechanisms underlying these findings by testing whether brain levels of primary BAs and gene expression of their principal receptors are altered in AD.

- Step 1: We assayed serum concentrations CA, CDCA, and 7α-OHC and used linear regression and mixed effects models to test their associations with brain amyloid accumulation ($N$ = 141), WMLs, and brain atrophy ($N$ = 134) in the Baltimore Longitudinal Study of Aging (BLSA). The BLSA is an ongoing, community-based cohort study that began in 1958. Participants in the BLSA neuroimaging sample were approximately 46% male with a mean age of 76 years; longitudinal analyses included an average of 2.5 follow-up magnetic resonance imaging (MRI) visits. We used the Alzheimer's Disease Neuroimaging Initiative (ADNI) ($N$ = 1,666) to validate longitudinal neuroimaging results in BLSA. ADNI is an ongoing, community-based cohort study that began in 2003. Participants were approximately 55% male with a mean age of 74 years; longitudinal analyses included an average of 5.2 follow-up MRI visits. Lower serum concentrations of 7α-OHC, CA, and CDCA were associated with higher brain amyloid deposition ($p$ = 0.041), faster WML accumulation ($p$ = 0.050), and faster brain atrophy mainly (false discovery rate [FDR] $p$ = <0.001–0.013) in males in BLSA. In ADNI, we found a modest sex-specific effect indicating that lower serum concentrations of CA and CDCA were associated with faster brain atrophy (FDR $p$ = 0.049) in males.

- Step 2: In the Clinical Practice Research Datalink (CPRD) dataset, covering >4 million registrants from general practice clinics in the United Kingdom, we tested whether patients using BAS (BAS users; 3,208 with ≥2 prescriptions), which reduce circulating BAs and increase cholesterol catabolism, had altered dementia risk compared to those on non-statin lipid-modifying therapies (LMT users; 23,483 with ≥2 prescriptions). Patients in the study (BAS/LMT) were approximately 34%/38% male and with a mean age of 65/68 years; follow-up time was 4.7/5.7 years. We found that BAS use was not significantly associated with risk of all-cause dementia (hazard ratio (HR) = 1.03, 95% confidence interval (CI) = 0.72–1.46, $p$ = 0.88) or its subtypes. We found a significant difference between the risk of VaD in males compared to females ($p$ = 0.040) and a significant dose–response relationship between BAS use and risk of VaD ($p$-trend = 0.045) in males.

- Step 3: We assayed brain tissue concentrations of CA and CDCA comparing AD and control (CON) samples in the BLSA autopsy cohort ($N$ = 29). Participants in the BLSA autopsy cohort (AD/CON) were approximately 50%/77% male with a mean age of 87/82 years. We analyzed single-cell RNA sequencing (scRNA-Seq) data to compare brain BA receptor gene expression between AD and CON samples from the Religious Orders Study and Memory and Aging Project (ROSMAP) cohort ($N$ = 46). ROSMAP is an ongoing, community-based cohort study that began in 1994. Participants (AD/CON) were approximately 56%/36% male with a mean age of 85/85 years. In BLSA, we found that CA and CDCA were detectable in postmortem brain tissue samples and were marginally higher in AD samples compared to CON. In ROSMAP, we found sex-specific differences in altered neuronal gene expression of BA receptors in AD. Study limitations include the small sample sizes in the BLSA cohort and likely inaccuracies in the clinical diagnosis of dementia subtypes in primary care settings.

EMR, electronic medical record; FDR, false discovery rate; FPR1, Formyl Peptide Receptor 1; GPBAR1, G Protein-Coupled Bile Acid Receptor 1; HNF4A, Hepatocyte Nuclear Factor 4 Alpha; HR, hazard ratio; IQR, interquartile range; IRB, Institutional Review Board; ISAC, Independent Scientific Advisory Committee; ITG, inferior temporal gyrus; KDR, Kinase Insert Domain Receptor; LMT, lipid-modifying therapies; LOD, limit of detection; MCI, mild cognitive impairment; MFG, middle frontal gyrus; MRI, magnetic resonance imaging; NIA, National Institute on Aging; NOS, not otherwise specified; NR1H2, Nuclear Receptor Subfamily 1 Group H Member 2; NR1H3, Nuclear Receptor Subfamily 1 Group H Member 3; NR1H4, Nuclear Receptor Subfamily 1 Group H Member 4; NR1I2, Nuclear Receptor Subfamily 1 Group I Member 2; NR1I3, Nuclear Receptor Subfamily 1 Group I Member 3; NR3C1, Nuclear Receptor Subfamily 3 Group C Member 1; NR5A2, Nuclear Receptor Subfamily 5 Group A Member 2; PET, positron emission tomography; PiB, [11]C-Pittsburgh compound-B; PPARA, Peroxisome Proliferator Activated Receptor Alpha; PPARD, Peroxisome Proliferator Activated Receptor Delta; PPARG, Peroxisome Proliferator Activated Receptor Gamma; RARA, Retinoic Acid Receptor Alpha; ROI, region of interest; ROSMAP, Religious Orders Study and Memory and Aging Project; RXRA, Retinoid X Receptor Alpha; RXRB, Retinoid X Receptor Beta; RXRG, Retinoid X Receptor Gamma; SABV, sex as a biological variable; scRNA-Seq, single-cell RNA sequencing; STROBE, Strengthening the Reporting of Observational studies in Epidemiology; ULOQ, upper limit of quantification; VaD, vascular dementia; VDR, Vitamin D Receptor; WML, white matter lesion.

## Conclusions

We combined targeted metabolomics in serum and amyloid positron emission tomography (PET) and MRI of the brain with pharmacoepidemiologic analysis to implicate dysregulation of cholesterol catabolism in dementia pathogenesis. We observed that lower serum BA concentration mainly in males is associated with neuroimaging markers of dementia, and pharmacological lowering of BA levels may be associated with higher risk of VaD in males. We hypothesize that dysregulation of BA signaling pathways in the brain may represent a plausible biologic mechanism underlying these results. Together, our observations suggest a novel mechanism relating abnormalities in cholesterol catabolism to risk of dementia.

## Author summary

### Why was this study done?

- Hypercholesterolemia is associated with increased risk of Alzheimer disease (AD) and vascular dementia (VaD).

- However, cholesterol is impermeable to the blood–brain barrier (BBB), and it is unclear how peripheral cholesterol mediates risk of dementia.

- While prior research has examined the relationship between de novo cholesterol biosynthesis and dementia, few studies have assessed the role of cholesterol catabolism and its principal breakdown products, oxysterols, and primary bile acids (BAs) in dementia.

### What did the researchers do and find?

- We examined the role of cholesterol catabolism in dementia pathogenesis by first testing the association between serum oxysterols and BAs and neuroimaging markers of dementia. We also tested whether exposure to bile acid sequestrants (BAS) was associated with risk of dementia in a large, real-world clinical dataset. Finally, we tested plausible mechanisms underlying these associations by examining whether primary BAs and mRNA of their receptors were altered in the brain in dementia.

- We found that lower serum levels of 7α-hydroxycholesterol (7α-OHC) and primary BAs were associated with higher brain amyloid deposition, faster WML accumulation, and faster brain atrophy mainly in males. Consistent with this finding, we observed a sex difference in the association between use of BAS and risk of VaD.

- We found that primary BAs were detectable in the brain, and levels of gene expression of BA receptors were altered in AD mainly in males.

### What do these findings mean?

- Our findings suggest that cholesterol catabolism and BA synthesis may impact dementia progression through sex-specific effects on signaling pathways in the brain.

- These results set the stage for experimental studies to test whether BA signaling in the brain may be a novel therapeutic target in dementia.

## Introduction

Accumulating evidence suggests that dementias such as Alzheimer disease (AD) and vascular dementia (VaD) may be the terminal consequences of metabolic abnormalities, such as hypercholesterolemia, manifesting several years prior to the onset of cognitive impairment and functional decline.

While hypercholesterolemia is associated with increased risk of both AD and VaD [1,2], the precise molecular mechanisms underlying this association remain unclear. Moreover, as cholesterol itself is impermeable to the blood–brain barrier (BBB), the question of how increased levels of peripheral cholesterol may mediate greater risk of dementia remains to be answered. These questions also bear important clinical translational implications for understanding how therapeutic targeting of cholesterol-related metabolic pathways may impact risk of dementia. The relationship between cholesterol metabolism and vascular disease has been predominantly studied from the perspective of de novo cholesterol biosynthesis, while relatively little attention has focused on cholesterol catabolism. The principal catabolic fate of cholesterol is its conversion to the primary bile acids (BAs), cholic acid (CA), and chenodeoxycholic acid (CDCA) through BBB-permeable intermediate metabolites called oxysterols (S1 Fig).

In order to test whether dysregulation of cholesterol catabolism is associated with dementia pathogenesis, we applied a 3-step study design.

First (Step 1), we used targeted metabolomics assays of serum samples within a longitudinal observational study to test whether serum concentrations of metabolites related to cholesterol catabolism, including the de novo synthesis of primary BAs, are associated with early neuroimaging markers of dementia including brain amyloid accumulation, white matter lesions (WMLs), and brain atrophy. We validated the association of BA levels with longitudinal neuroimaging outcomes in an independent cohort. Second (Step 2), based on associations between serum BA concentrations and neuroimaging markers of dementia identified in Step 1, we hypothesized that drugs targeting de novo BA synthesis, i.e., bile acid sequestrants (BAS) would alter risk of dementia. We tested this hypothesis in a large real-world clinical dataset. Third (Step 3), we explored plausible molecular mechanisms underlying our findings by testing whether levels of primary BAs and gene expression of their receptors were altered in the brain in dementia.

Given prior evidence suggesting sex-specific differences in the serum lipidome as well as in the association between lipid levels and dementia risk [3,4], we performed sex-stratified analyses to test the relationship between cholesterol catabolism and dementia.

## Materials and methods

Data used in our analyses were derived from the Baltimore Longitudinal Study of Aging (BLSA), the Alzheimer's Disease Neuroimaging Initiative (ADNI), the Alzheimer's Disease Metabolomics Consortium (ADMC), the Religious Orders Study and Memory and Aging Project (ROSMAP), and the Clinical Practice Research Datalink (CPRD). BLSA, ADNI, and ROSMAP are long-running, longitudinal cohorts established and prospectively followed to help address broad questions related to aging and disease. CPRD includes anonymized electronic medical record (EMR) data gathered from general practitioners in the United Kingdom. Specific analyses addressing focused hypotheses described herein were not included in prospective analysis plans in the original study protocols for these cohorts. BLSA, ADNI, and ROSMAP participants included in our analyses were a convenience sample available to researchers; power calculations to determine study size were not performed. Details on the analytic plan, including when specific plans were developed, are included in the Statistical

methods section below. This study is reported as per the Strengthening the Reporting of Observational studies in Epidemiology (STROBE) guidelines (S1 Table).

## Ethics approval

The BLSA study protocol has ongoing approval from the Institutional Review Board (IRB) of the National Institute of Environmental Health Science, National Institutes of Health ("Early Markers of Alzheimer's Disease (BLSA)", IRB No. 2009–074). Informed written consent was obtained at each visit from all participants.

The ADNI study protocol was approved by the IRBs of all the participating institutions/ study sites [5]. Informed written consent was obtained from all participants at each site. All ADNI studies are conducted according to Good Clinical Practice guidelines, the Declaration of Helsinki, and United States of America 21 CFR Part 50 (Protection of Human Subjects) and Part 56 (IRBs). Additional details can be found at adni.loni.usc.edu.

The ROSMAP study, including the parent study and substudies, was approved by the IRB of Rush University Medical Center. Informed written consent was obtained for all participants as well as an Anatomical Gift Act and a repository consent to share data and biospecimens.

CPRD data are anonymized, general practitioners do not need to seek patient consent when sharing data with CPRD, and patients have the option of opting out. Additional details can be found at https://www.cprd.com/public. This study was approved by the CPRD Independent Scientific Advisory Committee (ISAC; Protocol # Protocol 18_173) and exempted from full IRB review by the National Institutes of Health Office of Human Subject Research.

## Step 1: Test associations between cholesterol catabolism (i.e., BA synthesis) and neuroimaging markers of dementia

We performed targeted metabolomics assays measuring the principal cholesterol breakdown products (i.e., CA and CDCA) as well as their principal biosynthetic precursor, (7α-hydroxy-cholesterol; 7α-OHC) in serum samples from participants in the Baltimore Longitudinal Study of Aging Neuroimaging (BLSA-NI) substudy who also underwent in vivo brain amyloid positron emission tomography (PET) and longitudinal structural magnetic resonance imaging (MRI).

In order to validate index results from BLSA, we used the ADNI sample to test associations between CA and CDCA and neuroimaging outcomes (note: 7α-OHC was not assayed in the ADNI serum samples).

## Study participants

The BLSA is a prospective cohort study that began in 1958 and is administered by the National Institute on Aging (NIA) [6]. BLSA-NI substudy imaging and visit schedules have varied over time and have been described in detail previously [7] and included in S1 Text.

ADNI is an ongoing longitudinal study launched in 2003. The primary goal of ADNI has been to test whether longitudinal MRI, PET, and other biological markers can measure the progression of mild cognitive impairment (MCI) and early AD. ADMC performed serum BA assays in participants enrolled in ADNI, and data are publicly available. Study design details have been published previously [5] and are available at www.adni-info.org.

## Quantitative serum metabolomics assays

Blood serum samples were collected from BLSA participants at each visit; details on collection and processing have been published previously [8] and included in S1 Text.

Blood serum samples were collected from ADNI participants at baseline; details on collection and processing have been published previously [9,10] and are described in detail in the Biospecimen Results section (link: AD Metabolomics Consortium Bile Acids Methods (PDF)–Version: January 21, 2016) at http://adni.loni.usc.edu. Data used in this study are available in the Biospecimen Results section (link: AD Metabolomics Consortium Bile Acids–Post Processed Data [ADNI1, GO, 2]–Version: June 28, 2018).

### In vivo brain amyloid imaging, WMLs, and brain volumes

BLSA-NI participants underwent $^{11}$C-Pittsburgh compound-B (PiB) PET scans to assess brain amyloid-β burden. A detailed description of acquisition and preprocessing procedures has been published previously [11]. Individuals were characterized as amyloid +ve or amyloid −ve based on a mean cortical distribution volume ratio (cDVR) threshold of 1.066 [11]. Among amyloid +ve individuals, we examined mean cDVR, a weighted global average of brain amyloid deposition, and regional DVR in the precuneus, a region vulnerable to early amyloid deposition in AD [12]. The total sample included 141 individuals (66 male; 75 female) of whom 36 were amyloid +ve (21 male; 15 female).

BLSA brain MRI was performed on a 3T Philips Achieva scanner (Philips Healthcare, Netherlands) to quantify both global and regional brain volumes and WMLs. A detailed description is included in S1 Text.

We a priori defined a set of brain regions to examine brain atrophy over time based on prior work using BLSA-NI data suggesting that these regions were sensitive to age-related change [7]. These regions included global brain volumes: total brain, ventricular cerebrospinal fluid (CSF), total gray matter, and white matter; lobar volumes: temporal, parietal, and occipital white matter and gray matter; and additional regions sensitive to early neurodegeneration: hippocampus, entorhinal cortex, amygdala, parahippocampal gyrus, fusiform gyrus, and precuneus.

All BLSA MRI data, including brain volumes and WMLs, after onset of clinical symptoms among individuals who developed MCI or AD were excluded (21 visits). The total sample included 134 individuals (62 male; 72 female) with an average of 2.5 longitudinal MRI visits (male: 2.3; female: 2.6).

ADNI brain MRI was used to quantify both global and regional brain volumes. A detailed description of acquisition and preprocessing is included in S1 Text. As our analyses in the ADNI sample were performed to confirm index results from BLSA, we restricted these analyses to gray matter and subcortical brain regions described above, excluding all white matter regions based on the lack of associations in BLSA analyses.

ADNI is enriched for individuals with MCI and AD at baseline, and all data across baseline diagnoses (control (CON), MCI, and AD) were included in analyses. Similar to our primary analyses in BLSA, MRI data from all CON individuals in ADNI, after onset of clinical symptoms among individuals who subsequently developed MCI or AD, were excluded (200 visits). The total sample included 1,666 individuals (918 male; 748 female) with an average of 5.2 longitudinal MRI visits (male: 5.3; female: 5.1).

### Statistical methods

Step 1 of the analytic plan using BLSA data were developed in January 2018 prior to starting analyses in June 2019. The inclusion of ADNI BA and neuroimaging data to validate significant BLSA findings and sensitivity analyses (i.e., adding statin as a covariate) were performed in June 2020 in response to reviewer recommendations.

Similar to prior work in BLSA [8], metabolite concentrations above the upper limit of quantification (ULOQ) were excluded, concentrations below the limit of detection (LOD) were

imputed as the threshold LOD/2, and resulting concentrations were natural log transformed. Outliers ± 3 × interquartile range (IQR) were excluded.

For ADNI, data processing steps for serum BA concentrations have been described in detail previously [9,10]. Metabolite concentrations below the LOD were imputed as LOD/2, and all samples were $\log_2$ transformed. Outliers ± 3 × IQR were excluded.

To test for group differences between amyloid +ve and amyoid −ve individuals, we examined associations between serum concentrations of metabolites (i.e., 7α-OHC, CA, and CDCA) and brain amyloid deposition, in overall and sex-stratified linear regression models with metabolites as the dependent variable and the binary brain amyloid variable (i.e., amyloid +ve/amyloid −ve) as the main predictor. Covariates included mean-centered age and sex in the overall model and mean-centered age only in the sex-stratified model. We next tested the association between metabolite concentrations and mean cDVR (BLSA) and precuneus DVR (BLSA) in amyloid +ve individuals only. We used similar linear regression models for the continuous DVR predictors. The significance threshold was uncorrected and set at $p$ = 0.05 to accommodate the limited sample size.

To test the association between serum concentrations of metabolites (i.e., 7α-OHC, CA, and CDCA) and longitudinal changes in (1) regional brain volumes and (2) WMLs, we used total and sex-stratified linear mixed models with brain regions of interest (ROIs) volumes and WMLs as the dependent variable (i.e., outcome) and metabolite concentration as the predictor. We first performed analyses in BLSA and then validated results in ADNI. The statistical significance threshold for both BLSA and ADNI was set at a false discovery rate (FDR)-corrected $p$ = 0.05. Additional details on model specifications are included in S1 Text.

In sensitivity analyses, we explored the effect on associations of adding statin drug use as a covariate in BLSA models.

## Step 2: Test whether pharmacological modulation of BAs alters dementia risk in a large, real-world clinical dataset

In order to extend findings from our Step 1 analysis relating BA levels with brain amyloid accumulation, rates of brain atrophy, and progression of WMLs, we next tested whether pharmacological modulation of de novo BA synthesis influences dementia risk. As BAS are lipid-modifying treatments that are known to decrease the circulating pool of BAs [13–15] and promote the breakdown of cholesterol, we hypothesized that exposure to these drugs would alter the risk of dementia. We therefore tested associations between exposure to BAS and dementia risk using data from the UK's CPRD, an anonymized electronic health record (EHR) covering more than 4 million active registrants from the UK general practice clinics. Our results from Step 1 suggested a plausible sex difference in the effect of BAS on dementia risk, a hypothesis that we tested using the CPRD.

## Data source and study population

The CPRD is a primary care database covering >four million active registrants from >650 general practice clinics and is representative of the broader UK population in terms of age and sex [16].

From the August 2018 CPRD data release, we identified patients ≥18 years old who had a first prescription record (i.e., new users) for BAS (colestipol, colesevelam, and cholestyramine) or non-statin lipid-modifying therapies (LMT; fibrate, cholesterol absorption inhibitor, nicotinic acid derivative, and probucol) between January 1, 1995 and August 1, 2018. BAS are often used as a second-line treatment independently or in combination with statins, and therefore, we selected non-statin LMT users as an active comparator group. In both groups (BAS or

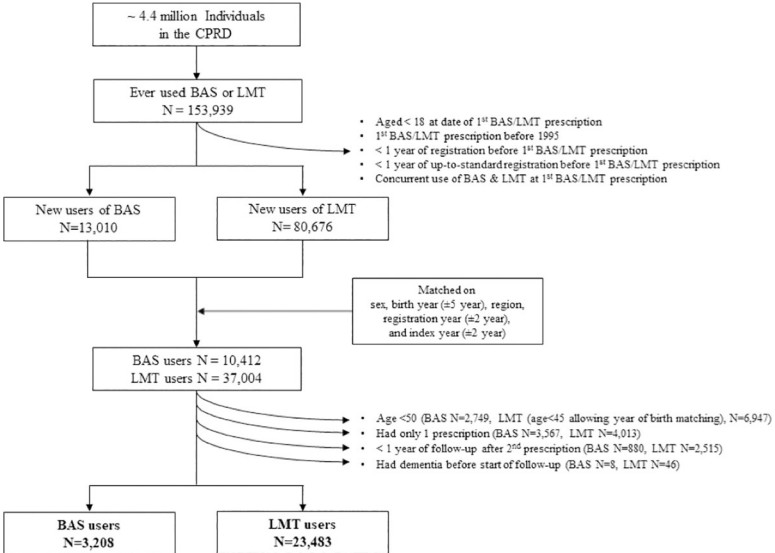

**Fig 1. Flowchart of study participants included in the CPRD analyses.** BAS, bile acid sequestrants including colestipol, colesevelam, and cholestyramine; CPRD, Clinical Practice Research Datalink; LMT, lipid-modifying therapies including fibrate, cholesterol absorption inhibitor, nicotinic acid derivative, and probucol.

non-statin LMTs), we allowed for prior statin use in combination with either BAS or LMTs. Individuals who only had a prescription record of statin use were excluded from this study.

The index date was defined as the date of the first BAS or LMT prescription. For each BAS user, we selected up to five LMT users matched on sex, year of birth (±5 year), region, year of clinic registration (±2 year), and year of first prescription (±2 years). Analysis was restricted to those with at least 12 months of clinical registration prior to the index date (to allow for covariate evaluation). We restricted BAS users to those aged ≥50 years and to those with two or more BAS/LMT prescriptions. The final analysis included 3,208 (1,083 male; 2,125 female) new BAS users and 23,483 (8,977 male; 14,506 female) new LMT users (Fig 1).

The outcomes of interest were all-cause dementia, and its subtypes: AD, VaD, and other dementia not otherwise specified (NOS). We used the last reported dementia diagnosis to identify the disease subtype (read codes are available upon request). The significance threshold was set at $p = 0.05$ considering that each outcome of interest was a priori specified.

## Statistical methods

Step 2 of the analytic plan using CPRD data was developed in January 2018 prior to starting data analyses in June 2019. We added a comparison of patient characteristics across outcome categories (i.e., dementia subtypes) based on reviewer recommendations.

We compared patient characteristics and their comorbidity profiles across dementia subtypes (AD, VaD, and NOS) as well as drug use (BAS and LMT) using the chi-squared test for categorical variables and Wilcoxon rank-sum tests for continuous variables.

For multivariable analyses, we used Cox proportional hazard models to calculate hazard ratios (HRs) and 95% confidence intervals (CIs) comparing dementia risk (all-cause and subtypes) in BAS versus LMT users in the overall and sex-stratified samples. Our sex-specific analyses were a priori specified and based on findings from Step 1. We also tested the dose–effect relationship between dementia risk and drugs of interest using the number of prescriptions. Models were adjusted for factors that were significantly different between BAS and LMT

groups to account for potential confounding by indication (since patient comorbidity profiles can lead to a BAS versus LMT prescription decision). Models were also adjusted for statin use during follow-up (until one year prior to exit date) using a time-varying covariate to account for its impact on dementia (26% versus 80% of BAS and LMT users, respectively, were prescribed statins in the 12 months before the index date and the majority (63%) continued its use for all or part of the follow-up). See S1 Text for details on model specifications.

## Step 3: Test plausible molecular mechanisms relating BA signaling in the brain to dementia pathogenesis using targeted metabolomics and transcriptomics

Given our findings that peripheral levels of the primary BAs, CA, and CDCA are associated with neuroimaging markers of dementia (Step 1) and that their pharmacological manipulation influences dementia risk (Step 2), we hypothesized that alterations in brain BA-mediated signaling may be a plausible biological mechanism underlying these findings. We first tested whether concentrations of CA and CDCA were detectable in the brain and whether they were altered in AD in participants from the BLSA autopsy program. We then tested whether gene expression of BA receptors was altered in AD using single-cell RNA sequencing (scRNA-Seq) data from the ROSMAP autopsy program.

### Study participants

The autopsy program of the BLSA was initiated in 1986 and has been described previously [17]. See S1 Text for additional details. Tissue samples from AD ($n$ = 16) and CON ($n$ = 13) from the inferior temporal gyrus (ITG) and middle frontal gyrus (MFG), regions representing areas of early neurofibrillary (i.e., tau) and neuritic plaque (i.e., amyloid) accumulation, respectively [18,19], as well as the cerebellum (CB) were included in these analyses.

scRNA-Seq gene expression data [20] from ROSMAP [21] were downloaded from Synapse (https://www.synapse.org/#!Synapse:syn18485175) under the doi 10.7303/syn18485175; code used to run analyses presented in Mathys and colleagues [20] was requested from coauthors. Data came from postmortem participants in ROSMAP including 46 individuals: 32 individuals (18 male and 14 female) in the AD category and 14 individuals (5 male and 9 female) in the CON category. The AD category included individuals with a clinical diagnosis of AD, including individuals with AD and no other condition contributing to cognitive impairment and AD and another condition contributing to cognitive impairment, as well as individuals with a clinical diagnosis of MCI and no other condition contributing to cognitive impairment. The CON category included individuals with a clinical diagnosis of no cognitive impairment.

Tissue was profiled from the prefrontal cortex (Brodmann area 10) across eight major cell types in the aged dorsolateral prefrontal cortex including inhibitory neurons, excitatory neurons, astrocytes, oligodendrocytes, microglia, oligodendrocyte progenitor cells, endothelial cells, and pericytes. Additional details are provided in the index paper [20]. We identified BA receptor genes (including receptors involved in BA homeostasis) using a literature search [22–24] and include the full list in S2 Table. There were 21 BA receptor genes that had available data in the ROSMAP dataset: Nuclear Receptor Subfamily 1 Group I Member 3 (NR1I3); Retinoid X Receptor Gamma (RXRG); Nuclear Receptor Subfamily 5 Group A Member 2 (NR5A2); Cholinergic Receptor Muscarinic 3 (CHRM3); G Protein-Coupled Bile Acid Receptor 1 (GPBAR1); Peroxisome Proliferator Activated Receptor Gamma (PPARG); Nuclear Receptor Subfamily 1 Group I Member 2 (NR1I2); Kinase Insert Domain Receptor (KDR); Nuclear Receptor Subfamily 3 Group C Member 1 (NR3C1); Retinoid X Receptor Beta (RXRB); Peroxisome Proliferator Activated Receptor Delta (PPARD); Cholinergic Receptor

Muscarinic 2 (CHRM2); Retinoid X Receptor Alpha (RXRA); Nuclear Receptor Subfamily 1 Group H Member 3 (NR1H3); Vitamin D Receptor (VDR); Nuclear Receptor Subfamily 1 Group H Member 4 (NR1H4); Retinoic Acid Receptor Alpha (RARA); Hepatocyte Nuclear Factor 4 Alpha (HNF4A); Nuclear Receptor Subfamily 1 Group H Member 2 (NR1H2); Formyl Peptide Receptor 1 (FPR1); Peroxisome Proliferator Activated Receptor Alpha (PPARA).

### Quantitative brain metabolomics assays

Quantitative metabolomics assays were performed on brain tissue samples to measure concentrations of the primary BAs, including CA and CDCA, using the Biocrates Bile Acids kit (Biocrates Life Sciences AG, Austria). Details on both assay kits, as well as calibration steps, have been published previously [25]. Additional details regarding the use of internal standards are included in S1 Text.

### Statistical methods

Step 3 of the analytic plan using BLSA data was developed in January 2020 in order to address a plausible molecular mechanism explaining findings from Step 1 and Step 2. The inclusion of scRNA-Seq data from ROSMAP occurred in June 2020 in response to reviewer recommendations to use non-array, non-bulk tissue-based gene expression data.

In order to assess whether primary BAs were present in the brain, we visualized CA and CDCA concentrations in AD and CON samples in the ITG, MFG, and CB using dot plots. Concentrations above and below LOD were indicated. We used tobit regression models to determine whether mean metabolite concentrations were significantly different between AD and CON samples. We set the lower limit as the metabolite-specific LOD threshold and included covariates age and sex (mean centered). In brain regions where metabolite concentrations were all below LOD (i.e., CB), we used chi-squared tests to determine whether percentage of samples below LOD was significantly different between AD and CON samples. Due to a small number of individuals with BA metabolite values above LOD, we were not able to sex-stratify these analyses. Additionally, the statistical significance threshold was set at $p = 0.05$ to accommodate the limited sample size.

For gene expression data, we scaled each sample to have the same total read count. To test differences between AD and CON, we used the Wilcoxon rank-sum test in the total and sex-stratified samples. Similar to the index paper [20], each single-cell–specific sample from a participant was treated as an independent sample. We summarized age- and sex-corrected fold changes (total sample) as well as sex-specific fold changes indicating whether genes were differentially expressed in AD versus CON samples. We additionally visualized results for significant associations using a heatmap. The significance threshold was set at an FDR-corrected $p = 0.05$.

## Results

### Step 1: Test associations between cholesterol catabolism (i.e., BA synthesis) and neuroimaging markers of dementia

Participant demographic details are included in S3A and S3B Table. Results of cross-sectional analyses testing associations between serum metabolite concentrations and amyloid status in BLSA are included in S4 Table, and associations between serum metabolite concentrations and brain amyloid-β burden among amyloid +ve individuals are included in Table 1. Brain amyloid +/−ve status was not significantly associated with serum concentrations of 7α-OHC, CDCA, or CA in the total or sex-stratified samples. In the total sample and in males only, serum 7α-OHC concentration, representing the rate-limiting biosynthetic precursor of the primary BAs [26,27], was significantly, negatively associated with mean cDVR ($p = 0.034$ and

**Table 1. Associations between serum metabolite concentrations and brain amyloid-β deposition, rates of brain atrophy, and longitudinal changes in global brain WML burden–BLSA.**

| | Global DVR (amyloid-β deposition) – amyloid +ve sample | | | | | |
|---|---|---|---|---|---|---|
| | Total (n = 36) | | Male (n = 21) | | Female (n = 15) | |
| | coef | pval | coef | pval | coef | pval |
| 7α-OHC | −1.439 | 0.034 | −1.568 | 0.041 | −1.007 | 0.57 |
| CDCA | 2.91 | 0.063 | 2.518 | 0.176 | 4.241 | 0.245 |
| CA | 3.198 | 0.05 | 3.209 | 0.104 | 2.993 | 0.446 |
| | **Precuneus DVR (amyloid-β deposition) – amyloid +ve sample** | | | | | |
| | Total (n = 36) | | Male (n = 21) | | Female (n = 15) | |
| | coef | pval | coef | pval | coef | pval |
| 7α-OHC | −1.197 | 0.033 | −1.447 | 0.022 | 0.09 | 0.948 |
| CDCA | 2.418 | 0.061 | 2.06 | 0.186 | 4.766 | 0.082 |
| CA | 1.966 | 0.152 | 1.994 | 0.236 | 2.062 | 0.501 |

| | Brain atrophy | | | | | | | | |
|---|---|---|---|---|---|---|---|---|---|
| | Total (n = 134) | | | Male (n = 62) | | | Female (n = 72) | | |
| | coef | pval | pval (FDR) | coef | pval | pval (FDR) | coef | pval | pval (FDR) |
| CDCA (Parietal GM) | 0.034 | 0.426 | 0.719 | 0.289 | <0.001 | 0.003 | −0.112 | 0.014 | 0.119 |
| CDCA (Precuneus) | 0.019 | 0.1 | 0.491 | 0.094 | <0.001 | <0.001 | −0.022 | 0.101 | 0.299 |
| CA (Parietal GM) | 0.022 | 0.577 | 0.797 | 0.207 | 0.001 | 0.013 | −0.154 | 0.001 | 0.038 |
| CA (Precuneus) | 0.016 | 0.139 | 0.602 | 0.069 | <0.001 | <0.001 | −0.031 | 0.02 | 0.119 |

| | WML | | | | | |
|---|---|---|---|---|---|---|
| | Total (n = 134) | | Male (n = 62) | | Female (n = 72) | |
| | coef | pval | coef | pval | coef | pval |
| 7α-OHC | 0.015 | 0.088 | −0.01 | 0.421 | 0.031 | 0.01 |
| CDCA | −0.001 | 0.892 | −0.009 | 0.05 | 0.005 | 0.342 |
| CA | −0.001 | 0.769 | −0.001 | 0.763 | 0 | 0.992 |

Significance threshold set at $p = 0.05$ for amyloid-β deposition and WMLs and FDR-corrected $p = 0.05$ for brain atrophy. Coefficients in green indicate that lower serum concentration of the metabolite is significantly associated with higher levels of brain amyloid-β, faster accumulation of WML accumulation, or faster brain atrophy. Coefficients in red indicate that lower serum concentration of the metabolite is significantly associated with lower levels of brain amyloid-β, slower accumulation of WML, or slower brain atrophy. Coef and pval highlighted in gray were not statistically significant.

7α-OHC, 7α-hydroxycholesterol; BLSA, Baltimore Longitudinal Study of Aging; CA, cholic acid; CDCA, chenodeoxycholic acid; cDVR, mean cortical DVR; coef, coefficient from linear regression model or mixed effects model; DVR, distribution volume ratio; FDR, false discovery rate (Benjamini–Hochberg) corrected $p$-value; GM, gray matter; pval, $p$-value; WML, white matter lesion.

$p = 0.041$, respectively) (Fig 2) and precuneus DVR ($p = 0.033$ and $p = 0.022$, respectively), indicating that lower serum concentration of 7α-OHC was associated with higher levels of global and precuneus brain amyloid-β deposition. We observed no significant associations in the female-only sample.

Results of longitudinal analyses in BLSA testing associations between serum metabolite concentrations and brain atrophy are shown in Table 1. In males, lower serum CDCA and CA concentrations were associated with faster rates of atrophy in the parietal gray matter (CDCA: FDR $p = 0.003$; CA: FDR $p = 0.013$) and precuneus (CDCA: FDR $p < 0.001$; CA: FDR $p < 0.001$). In females, lower serum CA concentration was associated with slower total gray matter atrophy (FDR $p = 0.038$).

Sensitivity analyses including statin drug use as a covariate are included in S5 Table; results were not substantially altered.

Results of longitudinal analyses in ADNI testing associations between serum metabolite concentrations and brain trophy are shown in Table 2. In the total sample, lower serum

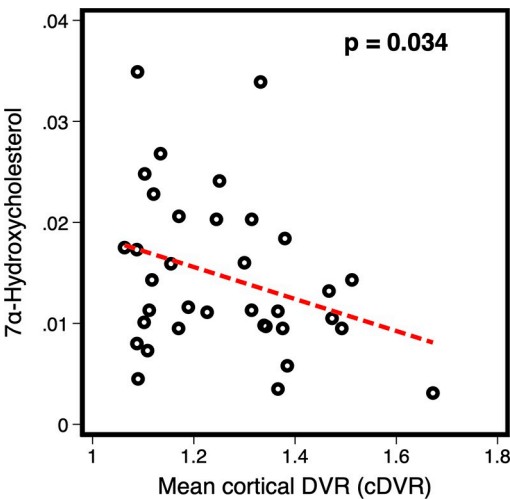

**Fig 2. Association between serum 7α-OHC concentration and brain amyloid-β burden in amyloid +ve individuals.** Weighted global average of brain amyloid-β burden (mean cDVR); analyses restricted to amyloid +ve individuals ($n = 36$). 7α-OHC, 7α-hydroxycholesterol; cDVR, cortical distribution volume ratio.

CDCA and CA concentrations were associated with faster rates of atrophy in the entorhinal cortex (CDCA: FDR $p = 0.032$; CA: FDR $p = 0.009$), frontal gray matter (CDCA: FDR $p = 0.045$; CA: FDR $p = 0.005$), fusiform gyrus (CDCA: FDR $p = 0.012$; CA: FDR $p = 0.001$), total gray matter (CDCA: FDR $p = 0.030$; CA: FDR $p = 0.003$), hippocampus (CDCA: FDR $p = 0.030$; CA: FDR $p = 0.012$), parahippocampal gyrus (CDCA: FDR $p = 0.012$; CA: FDR $p = 0.009$), temporal gray matter (CDCA: FDR $p = 0.016$; CA: FDR $p = 0.002$), and ventricles (CDCA: FDR $p = 0.030$; CA: FDR $p = 0.008$). Lower CA was also associated with faster rates of atrophy in the amygdala (FDR $p = 0.030$), occipital gray matter (FDR $p = 0.012$), parietal gray matter (FDR $p = 0.016$), and precuneus (FDR $p = 0.030$). In males, lower serum CDCA and CA concentrations were associated with faster rates of atrophy in the parahippocampal gyrus (CDCA: FDR $p = 0.049$; CA: FDR $p = 0.049$). Lower serum CDCA was associated with faster rates of atrophy in the ventricles (CDCA: FDR $p = 0.049$), and lower serum CA was associated with faster rates of atrophy in the entorhinal cortex, frontal gray matter, fusiform gyrus, total gray matter, hippocampus, and temporal gray matter (FDR $p = 0.049$). In females, CDCA was not significantly associated with rates of brain atrophy; lower CA was associated with faster rates of atrophy in only the fusiform gyrus and temporal gray matter (FDR $p = 0.039$).

Results of longitudinal analyses in BLSA testing associations between serum metabolite concentrations and WML are shown in Table 1. In males, lower serum CDCA concentration was associated with faster accumulation of WML ($p = 0.050$), and in females, lower serum 7α-OHC was associated with slower accumulation of WML ($p = 0.010$).

Results of longitudinal analyses in ADNI testing associations between serum metabolite concentrations and WML are shown in Table 2. In ADNI, we did not observe significant associations between serum metabolite concentrations of BAs (i.e., CA and CDCA) and WML in the total male or female samples.

## Step 2: Test whether pharmacological modulation of BAs alters dementia risk in a large, real-world clinical dataset

S6 Table summarizes characteristics of BAS and LMT users. LMT users were more likely to be overweight or obese compared with BAS users (73% versus 57%, respectively). In the 12

**Table 2. Associations between serum metabolite concentrations and rates of brain atrophy and longitudinal changes in global brain WML burden–ADNI.**

| | Brain atrophy | | | | | | | | |
|---|---|---|---|---|---|---|---|---|---|
| | Total (*n* = 1,666) | | | Male (*n* = 918) | | | Female (*n* = 748) | | |
| | coef | pval | pval (FDR) | coef | pval | pval (FDR) | coef | pval | pval (FDR) |
| CDCA (Ent. cortex) | 2.341 | 0.025 | 0.032 | 2.802 | 0.059 | 0.115 | 1.735 | 0.229 | 0.262 |
| CDCA (Frontal GM) | 46.678 | 0.037 | 0.045 | 45.613 | 0.126 | 0.191 | 52.38 | 0.122 | 0.177 |
| CDCA (Fus. gyrus) | 7.382 | 0.005 | 0.012 | 6.302 | 0.063 | 0.115 | 9.188 | 0.024 | 0.083 |
| CDCA (Total GM) | 152.971 | 0.022 | 0.03 | 124.115 | 0.159 | 0.201 | 191.975 | 0.06 | 0.12 |
| CDCA (Hipp) | 2.439 | 0.023 | 0.03 | 2.244 | 0.134 | 0.191 | 2.613 | 0.086 | 0.148 |
| CDCA (Parahipp. gyrus) | 3.262 | 0.005 | 0.012 | 3.841 | 0.016 | 0.049 | 2.574 | 0.125 | 0.177 |
| CDCA (Temporal GM) | 43.824 | 0.009 | 0.016 | 31.99 | 0.143 | 0.191 | 59.941 | 0.021 | 0.083 |
| CDCA (Ventricle) | −64.575 | 0.022 | 0.03 | −92.627 | 0.018 | 0.049 | −30.85 | 0.438 | 0.438 |
| CA (Amygdala) | 1.005 | 0.02 | 0.03 | 1.103 | 0.049 | 0.108 | 0.801 | 0.244 | 0.266 |
| CA (Ent. cortex) | 3.524 | 0.003 | 0.009 | 3.891 | 0.014 | 0.049 | 2.898 | 0.097 | 0.155 |
| CA (Frontal GM) | 83.879 | 0.001 | 0.005 | 85.173 | 0.007 | 0.049 | 85.187 | 0.037 | 0.098 |
| CA (Fus. gyrus) | 11.923 | <0.001 | 0.001 | 10.066 | 0.005 | 0.049 | 15.384 | 0.002 | 0.039 |
| CA (Total GM) | 264.396 | <0.001 | 0.003 | 231.124 | 0.013 | 0.049 | 320.791 | 0.009 | 0.055 |
| CA (Hipp) | 3.367 | 0.005 | 0.012 | 3.772 | 0.018 | 0.049 | 2.662 | 0.153 | 0.194 |
| CA (Occipital GM) | 31.538 | 0.005 | 0.012 | 21.78 | 0.137 | 0.191 | 47.546 | 0.008 | 0.055 |
| CA (Parahipp. gyrus) | 3.928 | 0.002 | 0.009 | 4.188 | 0.013 | 0.049 | 3.591 | 0.078 | 0.144 |
| CA (Parietal GM) | 35.172 | 0.008 | 0.016 | 26.732 | 0.106 | 0.181 | 48.186 | 0.029 | 0.088 |
| CA (Precuneus) | 8.589 | 0.022 | 0.03 | 6.543 | 0.178 | 0.203 | 11.897 | 0.044 | 0.105 |
| CA (Temporal GM) | 71.579 | <0.001 | 0.002 | 58.711 | 0.012 | 0.049 | 93.493 | 0.003 | 0.039 |
| CA (Ventricle) | −101.731 | 0.002 | 0.008 | −92.725 | 0.028 | 0.068 | −116.852 | 0.018 | 0.083 |
| | WML | | | | | | | | |
| | Total (*n* = 875) | | | Male (*n* = 456) | | | Female (*n* = 419) | | |
| | coef | pval | | coef | pval | | coef | pval | |
| CDCA | 11.714 | 0.326 | | 5.344 | 0.77 | | 18.075 | 0.216 | |
| CA | 17.236 | 0.215 | | 13.627 | 0.496 | | 23.858 | 0.198 | |

Coefficients in green indicate that lower serum concentration of the metabolite is significantly associated with faster accumulation of WML accumulation or faster brain atrophy. Coefficients in red indicate that lower serum concentration of the metabolite is significantly associated with slower accumulation of WML or slower brain atrophy. Coef and pval highlighted in gray were not statistically significant.

ADNI, Alzheimer's Disease Neuroimaging Initiative; CA, cholic acid; CDCA, chenodeoxycholic acid; coef, coefficient from linear regression model or mixed effects model; FDR, false discovery rate (Benjamini–Hochberg) corrected *p*-value. Significance threshold was set at an FDR-corrected *p* = 0.05; Fus. gyrus, fusiform gyrus; GM, gray matter; Hipp, Hippocampus; Parahipp gyrus, Parahippocampal gyrus; pval, *p*-value; WML, white matter lesion.

months prior to index date, LMT users were more likely to have used statins (80% versus 26%, respectively) or metformin (15% versus 7%) and had a record of coronary artery disease (7% versus 3%), type 2 diabetes (7% versus 3%), or dyslipidemia (25% versus 5%). BAS users were more likely to have a prior record of cancer (16% versus 8%).

Table 3 summarizes results from Cox proportional hazard models. During the median follow-up of 4.9 years, 809 incident dementia cases occurred (N = 72 for BAS versus 737 for LMT) corresponding to crude incidence rates of 4.8 (95% CI = 3.8 to 6.1) and 5.5 per 1,000 person-years (95% CI = 5.1 to 5.9) among BAS and LMT users, respectively. In multivariable adjusted models including all patients and compared to LMT use of > = two prescriptions, BAS use was not statistically significantly associated with risk of all-cause dementia (Table 3) or with its subtypes (any dementia: HR = 1.03, 95% CI = 0.72 to 1.46, *p* = 0.88; AD: HR = 1.24, 95% CI = 0.72 to 2.14, *p* = 0.43; VaD: HR = 1.27, 95% CI = 0.70 to 2.31, *p* = 0.43; other dementia: HR = 0.50, 95% CI = 0·22 to 1.15, *p* = 0.10).

**Table 3. Association between BAS use and the risk of dementia in individuals who received at least two BAS or LMT prescriptions with at least one year of follow-up after second prescription.**

| | | | | | Cause-specific analysis | | | | | | | |
| | | Any dementia | | | AD | | | Vascular disease | | | Other dementia, NOS[1] | | |
| Analysis | Exposure | N events/total | HR (95% CI)[2] | p | N events/total | HR (95% CI)[2] | p | N events/total | HR (95% CI)[2] | p | N events/total | HR (95% CI)[2] | p |
|---|---|---|---|---|---|---|---|---|---|---|---|---|---|
| **Overall** | | | | | | | | | | | | | |
| ≥2 RXs | BAS use[3] | 72/3,208 | **1.03 (0.72–1.46)** | **0.88** | 30/3,208 | **1.24 (0.72–2.14)** | **0.43** | 31/3,208 | **1.27 (0.70–2.31)** | **0.43** | 11/3,208 | **0.50 (0.22–1.15)** | **0.10** |
| | LMT use[3] | 737/23,483 | Reference | | 302/23,483 | Reference | | 260/23,483 | Reference | | 175/23,483 | Reference | |
| Cumulative BAS number of RXs | BAS 2 RXs | 16/810 | 0.95 (0.45–1.99) | | 5/810 | 0.56 (0.14–2.20) | | 7/810 | 2.12 (0.53–8.57) | | 4/810 | 0.70 (0.21–2.34) | |
| | BAS 3–5 RXs | 18/1,140 | 1.07 (0.50–2.30) | | 7/1,140 | 1.20 (0.39–3.69) | | 8/1,140 | 1.43 (0.40–5.09) | | 3/1,140 | 0.48 (0.05–4.49) | |
| | BAS ≥6 RXs | 38/1,744 | 1.04 (0.67–1.63) | | 18/1,744 | 1.60 (0.80–3.20) | | 16/1,744 | 1.09 (0.52–2.26) | | 4/1,744 | 0.37 (0.10–1.33) | |
| | LMT ≥2 RXs | 737/23,483 | Reference | | 302/23,483 | Reference | | 260/23,483 | Reference | | 175/23,483 | Reference | |
| | P for trend | | | 0.84 | | | 0.23 | | | 0.62 | | | 0.09 |
| **Male** | | | | | | | | | | | | | |
| ≥2 RXs | BAS use[4] | 25/1,083 | 1.20 (0.63–2.28) | 0.58 | 6/1,083 | 0.72 (0.19–2.78) | 0.64 | 14/1,083 | **2.89 (0.96–8.68)** | **0.06** | 5/1,083 | 0.52 (0.14–1.98) | 0.34 |
| | LMT use[4] | 240/8,977 | Reference | | 91/8,977 | Reference | | 90/8,977 | Reference | | 59/8,977 | Reference | |
| Cumulative BAS number of RXs | BAS 2 RXs | 7/261 | 1.74 (0.58–5.21) | | 2/261 | 4.03 (0.28–57.44) | | 2/261 | **1.43 (0.11–18.75)** | | 3/261 | 1.02 (0.19–5.38) | |
| | BAS 3–5 RXs | 4/367 | 0.52 (0.10–2.69) | | 1/367 | | | 3/367 | **2.41 (0.29–19.80)** | | 0/367 | | |
| | BAS ≥6 RXs | 14/591 | 1.25 (0.54–2.91) | | 3/591 | 0.69 (0.12–3.87) | | 9/591 | **3.85 (0.93–15.94)** | | 2/591 | 0.33 (0.03–3.16) | |
| | LMT ≥2 RXs | 240/8,977 | Reference | | 91/8,977 | Reference | | 90/8,977 | Reference | | 59/8,977 | Reference | |
| | P for trend | | | 0.71 | | | 0.47 | | | **0.045** | | | 0.23 |
| **Female** | | | | | | | | | | | | | |
| ≥2 RXs | BAS use[4] | 47/2,125 | 0.99 (0.65–1.52) | 0.98 | 24/2,125 | 1.35 (0.73–2.48) | 0.33 | 17/2,125 | 1.00 (0.46–2.15) | >0.99 | 6/2,125 | 0.43 (0.13–1.37) | 0.15 |
| | LMT use[4] | 497/14,506 | Reference | | 211/14,506 | Reference | | 170/14,506 | Reference | | 116/14,506 | Reference | |
| Cumulative BAS number of RXs | BAS 2 RXs | 9/549 | 0.64 (0.23–1.83) | | 3/549 | 0.17 (0.02–1.49) | | 5/549 | **2.95 (0.50–17.49)** | | 1/549 | 0.31 (0.03–3.23) | |
| | BAS 3–5 RXs | 14/773 | 1.53 (0.64–3.67) | | 6/773 | 2.05 (0.60–7.02) | | 5/773 | **1.42 (0.28–7.07)** | | 3/773 | 1.12 (0.11–11.64) | |
| | BAS ≥6 RXs | 24/1,153 | 0.97 (0.57–1.65) | | 15/1,153 | 1.85 (0.84–4.07) | | 7/1,153 | **0.69 (0.26–1.83)** | | 2/1,153 | 0.36 (0.07–1.83) | |
| | LMT ≥2 RXs | 497/14,506 | Reference | | 211/14,506 | Reference | | 170/14,506 | Reference | | 116/14,506 | Reference | |
| | P for trend | | | 0.93 | | | 0.12 | | | **0.66** | | | 0.18 |

1 Not otherwise specified.

2 Models were adjusted for smoking status, BMI, alcohol consumption close to index date, metformin use one year prior to index date, coronary artery diseases, type 2 diabetes and dyslipidemia record one year prior to index date, prior cancer history, time-varying statins use status during follow-up (until one year before exit date), and stratified on matched set.

3 ≥two RXs use.

Findings reported in the paper are indicated in **bold**.

AD, Alzheimer disease; BAS, bile acid sequestrants; BMI, body mass index; CI, confidence interval; HR, hazard ratio; LMT, non-statin lipid-modifying therapies; RX, prescription.

In analyses stratified by sex, we observed a significant ($p$ = 0.040) difference between the HR of VaD in males compared to females, indicating a sex difference in the relationship between BAS and risk of VaD. BAS use was associated with nonsignificantly elevated risk of VaD in males (HR = 2.89, 95% CI = 0.96 to 8.68, $p$ = 0.06). We identified a statistically significant dose–response relationship between BAS and risk of VaD in males. Specifically, risk of VaD was higher with the increased number of BAS prescriptions ($p$-trend = 0.045) (Table 3). There was no statistically significant association with VaD in females (overall or by number of prescriptions). Differences in patient characteristics across outcome categories are included in S7 Table.

## Step 3: Test plausible molecular mechanisms relating BA signaling in the brain to dementia pathogenesis using targeted metabolomics and transcriptomics

Participant demographic details for the BLSA autopsy study are included in S8 Table. Demographic details of ROSMAP participants included in scRNA-Seq analyses have been published previously [20].

The primary BAs, CDCA, and CA were detectable in postmortem brain tissue samples, although the majority were below the LOD (i.e., <LOD) (Fig 3, S9 Table). Tobit regression models indicated marginally higher (nonsignificant) concentrations of the primary BAs in AD

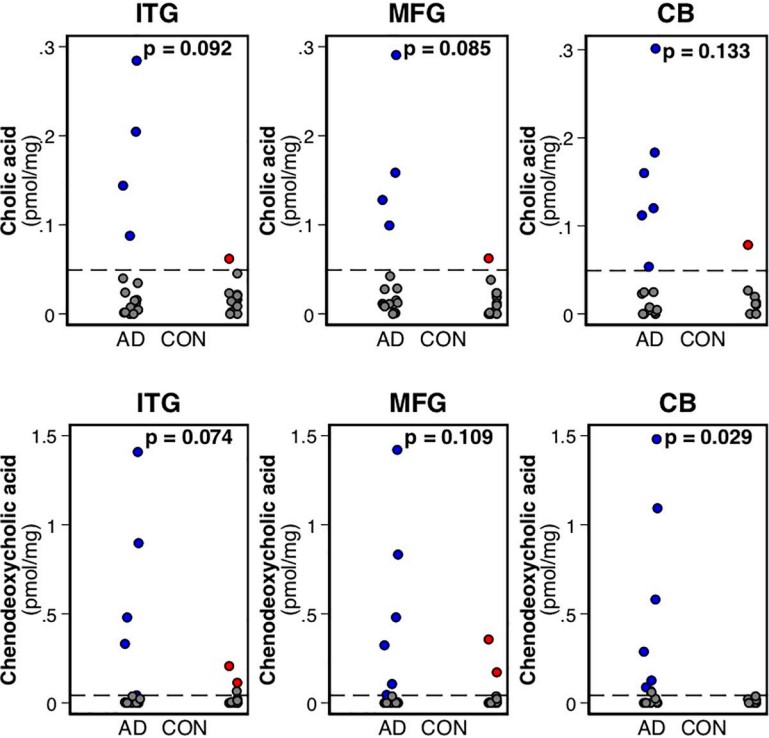

**Fig 3. Differences in brain primary BA concentrations between AD and CON.** Dots in gray indicate concentrations below the LOD; dots in blue and red indicate AD and CON brain tissue sample metabolite concentrations, respectively. $p$-Values indicate differences between AD and CON from a tobit regression model; the lower limit in the model is set at the metabolite-specific LOD (indicated by the dashed line). The $p$-value for CDCA in the CB is derived from a chi-squared test because there were no detectable concentrations above the LOD in the CON sample. AD, Alzheimer disease; BA, bile acid; CB, cerebellum; CDCA, chenodeoxycholic acid; CON, control; ITG, inferior temporal gyrus; LOD, limit of detection; MFG, middle frontal gyrus.

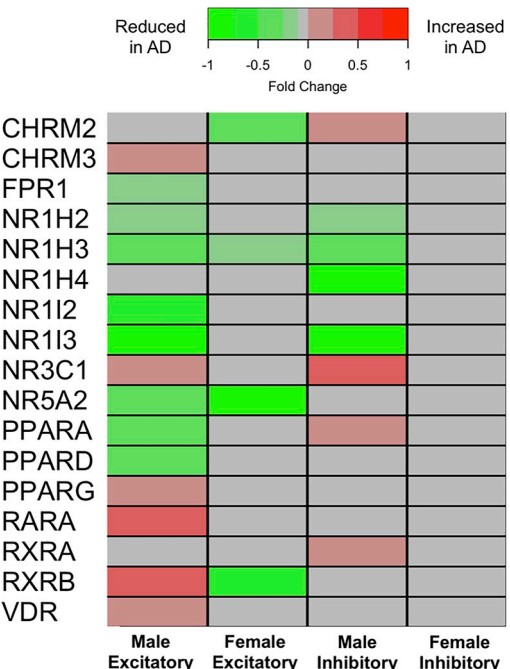

**Fig 4. Differences in brain BA receptor gene expression between AD and CON.** Summary of differentially expressed BA receptor genes (including receptors involved in BA homeostasis) in neurons in AD compared to CON. Statistically significant (FDR-corrected $p$-value < 0.05) fold change differences are indicated in green or red shading. Green shading indicates that gene expression was significantly reduced in AD compared to CON. Red shading indicates that gene expression was significantly increased in AD compared to CON. Gray shading indicates gene expression was not significantly different between AD and CON. AD, Alzheimer disease; BA, bile acid; CHRM2, Cholinergic Receptor Muscarinic 2; CHRM3, Cholinergic Receptor Muscarinic 3; CON, control; FDR, false discovery rate; FPR1, Formyl Peptide Receptor 1; NR1H2, Nuclear Receptor Subfamily 1 Group H Member 2; NR1H3, Nuclear Receptor Subfamily 1 Group H Member 3; NR1H4, Nuclear Receptor Subfamily 1 Group H Member 4; NR1I2, Nuclear Receptor Subfamily 1 Group I Member 2; NR1I3, Nuclear Receptor Subfamily 1 Group I Member 3; NR3C1, Nuclear Receptor Subfamily 3 Group C Member 1; NR5A2, Nuclear Receptor Subfamily 5 Group A Member 2; PPARA, Peroxisome Proliferator Activated Receptor Alpha; PPARD, Peroxisome Proliferator Activated Receptor Delta; PPARG, Peroxisome Proliferator Activated Receptor Gamma; RARA, Retinoic Acid Receptor Alpha; RXRA, Retinoid X Receptor Alpha; RXRB, Retinoid X Receptor Beta; VDR, Vitamin D Receptor.

samples compared to CON samples in the ITG and MFG. Chi-squared models indicated significantly more participants with metabolite concentrations above LOD in AD compared to CON in the CB. Due to a small number of individuals with BA metabolite values above LOD, we were not able to sex-stratify these analyses.

We observed that gene expression of several BA receptors was different in AD versus CON in the total and male samples, mainly within neurons (i.e., both inhibitory and excitatory neurons). The majority of differentially expressed genes showed lower expression in AD relative to CON. Results across all 8 major brain cell types are included in S10 Table. As indicated below in a heatmap visualizing sex-stratified differences in AD versus CON samples (Fig 4), for inhibitory neurons, in males, 10 out of 21 genes were significantly altered (FDR pval < 0.05); six had lower gene expression in AD compared to CON (AD<CON); and four had higher gene expression in AD compared to CON (AD>CON). In females, there were no differentially expressed BA receptor genes within inhibitory neurons. Within excitatory neurons, in males, 16 out of 21 genes were significantly altered (FDR pval < 0.05); 10 had lower gene expression in AD compared to CON (AD<CON); and 6 had higher gene expression in AD compared to CON (AD>CON). For females, four genes showed lower gene expression in AD compared to CON (AD<CON).

## Discussion

We found that lower serum concentrations of the rate-limiting biosynthetic precursor of BA synthesis, i.e., 7α-OHC, as well as the primary BAs mainly in males, were associated with neuroimaging measures of dementia progression and that pharmacological lowering of BA levels was associated with higher risk of VaD in males. We hypothesize that disruption of BA signaling in the brain as reflected in altered levels of primary BAs and reduced neuronal gene expression of BA receptors may represent a plausible biological mechanism underlying these results. Together, our observations suggest a novel mechanism relating abnormalities in cholesterol catabolism to risk of dementia.

The role of hypercholesterolemia in the pathogenesis of dementia is well recognized but poorly understood. While the BBB ensures that brain concentrations of cholesterol are largely independent of peripheral tissues [28], the oxidative catabolism of cholesterol results in the generation of oxysterols that are permeable to the BBB and can both access the brain from the peripheral circulation, as well as efflux into the periphery from the brain (S1 Fig). Oxysterols, including 7α-OHC, are key biosynthetic precursors of the primary BAs, CA, and CDCA, which, in turn, represent the primary catabolic products of cholesterol.

We observed an association between serum concentration of 7α-OHC, representing the rate-limiting reaction in primary BA synthesis [29–31], and global brain amyloid burden as well as that in the precuneus, an early site of amyloid deposition in AD [12] suggesting that impaired synthesis of primary BAs may be an important mediator of pathologic changes in AD. This relationship appears to be driven primarily by males suggesting a novel sex-specific association between BA synthesis and brain amyloid accumulation. It is important to note, however, that these cross-sectional analyses are not able to determine whether pathology, brain atrophy, or other dementia-associated endophenotypes may modify cholesterol catabolism.

We then examined the relationships between BA synthesis and both regional rates of brain atrophy as well as the accumulation of WMLs that are key vascular contributors to dementia [32]. Our results indicate that in males, lower serum CDCA and CA is associated with faster rates of brain atrophy and faster accumulation of brain WMLs in the BLSA. These findings were partially confirmed in ADNI where lower BA concentrations were associated with faster brain atrophy rates across several brain regions in males with far fewer associations in females. It is important to note, however, that the lack of sex-specific associations compared to the total sample in ADNI may be partially driven by sample size. Female participants in BLSA showed an opposite effect compared to males—lower serum concentrations of 7α-OHC and CA were associated with slower accumulation of brain WMLs and slower rates of brain atrophy. Our sex-specific WML associations in BLSA were not replicated in ADNI. This may be due, in part, to demographic differences: ADNI participants were younger at baseline and had a larger percentage of participants who were white. Additionally, ADNI participants represent later stages of disease progression compared to the BLSA sample with approximately 50% of participants at baseline being diagnosed as either MCI or AD.

To the best of our knowledge, these findings are among the first to demonstrate sex-specific associations between the rate-limiting step in primary BA synthesis and brain amyloid deposition as well as longitudinal changes in brain atrophy and accumulation of WML burden. A previous cross-sectional study by Nho and colleagues [9] in ADNI reported that lower plasma CA was associated with reduced hippocampal volume in a combined sample of AD, MCI, and CON participants and reported lower plasma CA levels in AD as well associations with increased risk of conversion from MCI to AD. These results, together with our current findings which included longitudinal markers of disease progression, suggest that the oxidative

catabolism of cholesterol to BAs may impact both pathological changes in the brain preceding a diagnosis of dementia, as well as progression of clinical symptoms [33].

Given that our neuroimaging results revealed sex-specific associations between primary BA synthesis and measures of dementia-related pathology, we next tested whether the modulation of peripheral BA levels would alter the risk of incident dementia in a sex-specific manner. To test this hypothesis, we leveraged one of the world's largest databases of primary care records, i.e., the UK CPRD, to examine whether exposure to BAS, a commonly used class of medicines to treat hyperlipidemia, would alter the risk of incident dementia. BAS are nonsystemic pharmacological agents that bind to BAs in the gastrointestinal tract, reducing their entry into the enterohepatic circulation. A lower pool of circulating BAs reduces feedback inhibition of the rate-limiting step in BA synthesis catalyzed by CYP7A1 [34], resulting in greater oxidative catabolism of cholesterol. We observed a significant positive association between the number of BAS prescriptions and risk of VaD in males and no association in females. We additionally observed a statistically significant sex difference in the association between BAS and VaD. These results, while suggestive, are consistent with our neuroimaging findings indicating that a lower circulating pool of BA is associated with neuroimaging markers of dementia progression mainly in males. Together, these results suggest that cholesterol catabolism through its enzymatic conversion to primary BAs is a biological mechanism associated with increased risk of VaD in males. These findings may provide novel insights into sex-specific interventions targeting this biochemical pathway in at-risk older individuals. Further exploration of the association between pharmacologic manipulation of BA levels and dementia outcomes in complementary population-based databases with distinct demographic and clinical characteristics is essential to validate our findings and assess their generalizability.

One plausible mechanism that may explain the association between dysregulated cholesterol catabolism and dementia pathogenesis is through altered BA signaling in the brain. Our findings are among the first to identify primary BAs (i.e., CA and CDCA) in the brain and report significant sex differences in neuronal gene expression of BA receptors in AD. A recent report by Mahmoudiandehkordi and colleagues reported a significant association between a higher ratio of the secondary BA, deoxycholic acid (DCA) to CA (DCA:CA) in both serum and brain tissue with severity of cognitive impairment in a combined sample of AD, MCI, and CON participants from the ROSMAP study [33]. Our scRNA-Seq results are also broadly consistent with a recent multi-cohort transcriptomic analysis in AD and CON brain tissue samples by Baloni and colleagues [22] that reported gene expression of several BA receptors. These included transcripts for RARA, RXRA, PPARA, and PPARG receptors that we find to be differentially expressed in AD brains in a sex-specific manner (Fig 4). Important differences between our current report and that by Baloni and colleagues include our use of scRNA-Seq compared to bulk tissue RNA-Seq as well as our sex-stratified analyses to probe differences in BA receptor transcript levels in AD. While the influx of BAs across the BBB from systemic circulation has been demonstrated [35], it is unclear whether de novo synthesis contributes substantially to the BA pool in the human brain [36]. Few previous studies have reported the existence of BAs in the human brain [37,38]. Pan and colleagues reported the presence of CDCA and CA in postmortem AD and CON brains but did not observe differences in their concentrations [37].

While BA receptors play a critical role in regulating hepatic BA synthesis by mediating feedback inhibition of CYP7A1, accumulating evidence also points to their importance in signaling pathways in the brain [39]. These include regulation of vascular risk factors including glucose, lipid, and energy homeostasis [40] as well as modulation of GABAergic and NMDA receptor–mediated neurotransmission [41]. Our results raise the possibility that dysregulation of cholesterol catabolism and BA synthesis in the periphery may impact early features of dementia

pathogenesis through their effects on neuronal signaling pathways in the brain. This hypothesis merits evaluation in future experimental studies and may pave the way toward testing novel disease-modifying treatments in dementia targeting BA receptor–mediated signaling in the brain.

While we have not addressed the precise mechanisms underlying sex-specific associations between BA metabolism and dementia pathogenesis, prior evidence suggests important sex differences in lipid metabolism that impact risk of cardiovascular disease [42,43]. It is important to consider these findings together with animal studies that have also shown sex–specific differences in BA homeostasis during aging and suggest that these may be mediated by differences in expression of BA transporters as well as CYP7A1, the rate-limiting enzyme in BA synthesis [44]. It is likely that such differences are relevant in other biological pathways as well. Our own prior work has uncovered striking sex differences in the systemic inflammatory response in preclinical AD that is related to neurodegeneration [45] as well as differences in glucose metabolism that are associated with AD pathology [46]. These findings may have implications for testing targeted treatment interventions that take into consideration sex-specific differences in molecular mechanisms underlying AD pathogenesis. Understanding sex differences in biological pathways related to AD risk and progression may also have important implications in our understanding of descriptive epidemiological estimates of dementia that reveal sex-specific longitudinal differences in both prevalence and incidence of dementia in diverse cohorts [47]. It is also worth noting in this context that sex as a biological variable (SABV) has been largely ignored in neuroscience and dementia research [48,49].

Our study design represents an approach to identify biological mechanisms of risk associated with dementia as well as to discover potential targets for disease-modifying treatments. First, the use of targeted metabolomics and transcriptomics within longitudinal observational studies in combination with established neuroimaging markers of disease progression (e.g., amyloid accumulation, brain atrophy, and WMLs) enables the identification of specific biochemical pathways that may present plausible drug targets. Second, the use of large, real-world clinical datasets with dementia outcomes enables testing drugs that may impact such targets.

The strengths of our study include the use of a well-characterized population of older individuals with serial neuroimaging in the BLSA-NI and ADNI and testing the clinical implications of our findings in a large real-world clinical dataset.

Limitations of our study include the relatively small sample sizes in the BLSA-NI and autopsy samples. Additionally, we were unable to sex-stratify analyses of brain tissue BA concentration due to a limited number of individuals with BA metabolite concentration values above LOD. However, our inclusion of sc-RNASeq data comparing AD and CON samples from ROSMAP did allow us to sex-stratify gene expression analyses and correct for multiple comparisons. Additional limitations include a likely inaccuracy in clinical diagnoses of dementia subtypes in primary care settings. We have previously analyzed data from more than 20 million Medicare fee-for-service beneficiaries in the USA and reported that accurate subtyping of dementia in such datasets may be challenging [50]. While our matched cohort design with an active drug comparator group and adjustment for common comorbidities may have addressed some of the limitations associated with pharmacoepidemiologic analyses, our findings merit confirmation in other independent studies. It is also important to note that large longitudinal studies have consistently reported that mixed brain pathologies account for the majority of dementia cases with considerable overlap between AD neuropathology and vascular brain injury including macroscopic, lacunar, and microscopic infarcts [51,52]. Additionally, particularly in the oldest old, "single neuropathological entities" [53] may be less relevant compared to mixed pathologies including AD and vascular disease [54–56].

In summary, we have combined targeted metabolomic assays of serum with in vivo amyloid PET and MRI of the brain to identify cholesterol catabolism through BA synthesis as a

biological pathway involved in neuropathological changes prior to dementia onset. We then extended these findings by analyzing a large real-world clinical dataset to show that BA modulation alters the trajectory of VaD in males. Our transcriptomics results suggest that alterations in BA signaling through their neuronal receptors may mediate some of these associations. Our findings suggest that future experimental studies may provide insight into modulation of BA levels as a plausible therapeutic target in dementia.

## Supporting information

**S1 Fig. Catabolism of cholesterol into primary BAs.** The oxidative catabolism of cholesterol occurs through 3 enzymatically catalyzed biochemical pathways: the classic/neutral pathway in the liver accounts for the majority of BA synthesis in humans and begins with the oxidation of cholesterol to 7α-OHC by microsomal CYP7A1, the rate-limiting enzyme of the pathway. The alternative or acidic pathway is responsible for synthesis of a smaller proportion of the BA pool; cholesterol is oxidized to 27-OHC, catalyzed by mitochondrial CYP27A1 in both liver and extra-hepatic tissues. Both the classic/neutral and acidic pathways of BA synthesis ultimately generate the primary BAs, CA, and CDCA which are the principal catabolic products of cholesterol. A third, neuron-specific pathway of cholesterol breakdown in the brain is catalyzed by CYP46A1-mediated conversion of cholesterol to 24S-OHC which effluxes into the peripheral circulation for further conversion into the primary BAs in the liver [31,40,57]. 7α-OHC, 7α-hydroxycholesterol; 24S-OHC, 24S-hydroxycholesterol; 27-OHC, 27-hydroxycholesterol; BA, bile acid; CA, cholic acid; CDCA, chenodeoxycholic acid.
(TIFF)

**S1 Text. Supporting information text.**
(DOCX)

**S1 Table. STROBE checklist.** STROBE, Strengthening the Reporting of Observational studies in Epidemiology.
(DOCX)

**S2 Table. ROSMAP scRNA-Seq BA receptor gene expression data availability.** Indicates data availability in the scRNA-Seq ROSMAP dataset. BA receptor genes that are indicated as "Not Available" either did not have sufficient counts or did not have any data available in the ROSMAP scRNA-Seq dataset. BA, bile acid; CHRM2, Cholinergic Receptor Muscarinic 2; CHRM3, Cholinergic Receptor Muscarinic 3; FGF19, Fibroblast Growth Factor 19; FPR1, Formyl Peptide Receptor 1; GPBAR1, G Protein-Coupled Bile Acid Receptor 1; HNF4A, Hepatocyte Nuclear Factor 4 Alpha; KDR, Kinase Insert Domain Receptor; NR0B2, Nuclear Receptor Subfamily 0 Group B Member 2; NR1H2, Nuclear Receptor Subfamily 1 Group H Member 2; NR1H3, Nuclear Receptor Subfamily 1 Group H Member 3; NR1H4, Nuclear Receptor Subfamily 1 Group H Member 4; NR1I2, Nuclear Receptor Subfamily 1 Group I Member 2; NR1I3, Nuclear Receptor Subfamily 1 Group I Member 3; NR3C1, Nuclear Receptor Subfamily 3 Group C Member 1; NR5A2, Nuclear Receptor Subfamily 5 Group A Member 2; PPARA, Peroxisome Proliferator Activated Receptor Alpha; PPARD, Peroxisome Proliferator Activated Receptor Delta; PPARG, Peroxisome Proliferator Activated Receptor Gamma; RARA, Retinoic Acid Receptor Alpha; ROSMAP, Religious Orders Study and Memory and Aging Project; RXRA, Retinoid X Receptor Alpha; RXRB, Retinoid X Receptor Beta; RXRG, Retinoid X Receptor Gamma; S1PR2, Sphingosine-1-Phosphate Receptor 2; scRNA-Seq, single-cell RNA sequencing; VDR, Vitamin D Receptor.
(DOCX)

**S3 Table.** **(A)** Demographic characteristics of BLSA-NI sample. APOE4, e4 allele of the Apolipoprotein E gene; BLSA, Baltimore Longitudinal Study of Aging; MRI, magnetic resonance imaging; NI, neuroimaging; PiB, Pittsburgh compound B; SD, standard deviation; WML, white matter lesion. **(B)** Demographic characteristics of ADNI sample. ADNI, Alzheimer's Disease Neuroimaging Initiative; MRI, magnetic resonance imaging; NI, neuroimaging; SD, standard deviation; WML, white matter lesion
(DOCX)

**S4 Table. Associations between serum metabolite concentrations and PiB/amyloid status.** coef, coefficient from linear regression model; PiB, Pittsburgh compound B; pval, *p*-value.
(DOCX)

**S5 Table. Sensitivity analyses: Associations between serum metabolite concentrations and brain amyloid-β deposition, longitudinal changes in global brain WML burden, and rates of brain atrophy–BLSA.** Sensitivity analyses after including statin use as a covariate. BLSA, Baltimore Longitudinal Study of Aging; coef, coefficient from linear regression model or mixed effects model; FDR, false discovery rate (Benjamini–Hochberg) corrected *p*-value; pval, *p*-value; WML, white matter lesion.
(DOCX)

**S6 Table. Characteristics of participants who received at least 2 BAS or LMT prescriptions with at least 1 year of follow-up after second prescription.** Wilcoxon rank-sum test. [1] Chi-squared test. [2] 1 year prior to index date. [3] During study follow-up. BAS, bile acid sequestrants; LMT, lipid-modifying therapies.
(DOCX)

**S7 Table. Characteristics of participants with incident dementia events during follow-up.** [1] 1 year prior to index date.
(DOCX)

**S8 Table. Demographic characteristics of BLSA autopsy sample.** AD, Alzheimer disease; APOE4, apolipoprotein E allele epsilon 4; BLSA, Baltimore Longitudinal Study of Aging; CON, control; PMI, postmortem interval (hours).
(DOCX)

**S9 Table. Differences in brain primary BA concentrations between AD and CON.** [*] In the CON sample in the CB, all concentrations were below LOD; we therefore tested for differences in the number of concentrations below LOD comparing AD to CON using the chi-squared test and present the associated *p*-value. AD, Alzheimer disease; BA, bile acid; coef, coefficient for disease (AD vs. CON) from the tobit model including mean-centered age and sex where the lower limit is set as the metabolite specific LOD; CON, control; LOD, limit of detection; pval, *p*-value.
(DOCX)

**S10 Table. Differences in brain BA receptor gene expression (including receptors involved in BA homeostasis) in AD compared to CON.** AD, Alzheimer disease; BA, bile acid; CHRM2, Cholinergic Receptor Muscarinic 2; CHRM3, Cholinergic Receptor Muscarinic 3; CON, control; FDR, false discovery rate (Benjamini–Hochberg) corrected *p*-value; FPR1, Formyl Peptide Receptor 1; GPBAR1, G Protein-Coupled Bile Acid Receptor 1; HNF4A, Hepatocyte Nuclear Factor 4 Alpha; KDR, Kinase Insert Domain Receptor; NR1H2, Nuclear Receptor Subfamily 1 Group H Member 2; NR1H3, Nuclear Receptor Subfamily 1 Group H Member 3; NR1H4, Nuclear Receptor Subfamily 1 Group H Member 4; NR1I2, Nuclear Receptor

Subfamily 1 Group I Member 2; NR1I3, Nuclear Receptor Subfamily 1 Group I Member 3; NR3C1, Nuclear Receptor Subfamily 3 Group C Member 1; NR5A2, Nuclear Receptor Subfamily 5 Group A Member 2; PPARA, Peroxisome Proliferator Activated Receptor Alpha; PPARD, Peroxisome Proliferator Activated Receptor Delta; PPARG, Peroxisome Proliferator Activated Receptor Gamma; pval: *p*-value; RARA, Retinoic Acid Receptor Alpha; RXRA, Retinoid X Receptor Alpha; RXRB, Retinoid X Receptor Beta; RXRG, Retinoid X Receptor Gamma; VDR, Vitamin D Receptor.
(XLSX)

## Acknowledgments

We are grateful to BLSA and ROSMAP participants for their invaluable contributions. The authors thank Emily Carver, BS, and David Ruggieri, BS, both from Information Management Services (Calverton, Maryland, United States of America), for their important contributions to database management. We would like to additionally thank Dr. Li-Huei Tsai, Dr. Hansruedi Mathys, Dr. Manolis Kellis, and Dr. Jose Davila Valderrain for their support with scRNA-Seq data acquisition and analysis. We are thankful for support from the ADMC and the ADNI studies. Investigators within the ADMC and ADNI contributed to the design and implementation of ADMC/ADNI and/or provided data but did not participate in the analysis or writing of this report. A complete listing of ADNI investigators can be found at http://adni.loni.usc.edu/wp-content/uploads/how_to_apply/ADNI_Authorship_List.pdf. A complete listing of ADMC investigators can be found at https://sites.duke.edu/adnimetab/team/.

**Disclaimers:** This study is based on data from the CPRD GOLD database August 2018 release, obtained under license from the UK Medicines and Healthcare Products Regulatory Agency. All rights reserved. The interpretation and conclusions contained in this study are those of the authors alone.

## Author Contributions

**Conceptualization:** Vijay R. Varma, Madhav Thambisetty.

**Data curation:** Vijay R. Varma, Youjin Wang, Sudhir Varma, Murat Bilgel, Jimit Doshi, Jackson A. Roberts, Dean F. Wong, Christos Davatzikos, Susan M. Resnick, Juan C. Troncoso, Olga Pletnikova, Ruth Pfeiffer, Priyanka Baloni, Siamak Mohmoudiandehkordi, Kwangsik Nho, Shahinaz M. Gadalla.

**Formal analysis:** Vijay R. Varma, Youjin Wang, Yang An, Sudhir Varma, Ruth Pfeiffer, Shahinaz M. Gadalla.

**Investigation:** Vijay R. Varma, Cristina Legido-Quigley, Christos Davatzikos, Susan M. Resnick, Juan C. Troncoso, Richard O'Brien, Shahinaz M. Gadalla, Madhav Thambisetty.

**Methodology:** Vijay R. Varma, Youjin Wang, Yang An, Murat Bilgel, Jimit Doshi, Cristina Legido-Quigley, Anup M. Oommen, Dean F. Wong, Susan M. Resnick, Olga Pletnikova, Ruth Pfeiffer, Madhav Thambisetty.

**Project administration:** Vijay R. Varma, Madhav Thambisetty.

**Resources:** Madhav Thambisetty.

**Supervision:** Vijay R. Varma, Madhav Thambisetty.

**Visualization:** Vijay R. Varma, Sudhir Varma, Jackson A. Roberts.

**Writing – original draft:** Vijay R. Varma, Youjin Wang, Madhav Thambisetty.

**Writing – review & editing:** Vijay R. Varma, Yang An, Murat Bilgel, Cristina Legido-Quigley, João C. Delgado, Anup M. Oommen, Christos Davatzikos, Susan M. Resnick, Eelko Hak, Brenda N. Baak, Ruth Pfeiffer, Priyanka Baloni, Rima Kaddurah-Daouk, David A. Bennett, Shahinaz M. Gadalla, Madhav Thambisetty.

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
