## [Editor Report · Decision Letter 0]

13 Feb 2020

Dear Dr Varma, 

Thank you for submitting your manuscript entitled "Bile acid synthesis and modulation are associated with brain amyloid deposition, white matter lesions, neurodegeneration and risk of vascular dementia" for consideration by PLOS Medicine.

Your manuscript has now been evaluated by the PLOS Medicine editorial staff and I am writing to let you know that we would like to send your submission out for external peer review.

Please re-submit your manuscript within two working days, i.e. by 17th Feb 2020, 11:59PM

Kind regards,

Louise Gaynor-Brook, MBBS PhD

Associate Editor, PLOS Medicine

---

## [Decision Letter · Decision Letter 1]

20 Jun 2020

Dear Dr. Varma,

Thank you very much for submitting your manuscript "Bile acid synthesis and modulation are associated with brain amyloid deposition, white matter lesions, neurodegeneration and risk of vascular dementia" (PMEDICINE-D-20-00371R1) for consideration at PLOS Medicine. 

[LINK]

In light of these reviews, I am afraid that we will not be able to accept the manuscript for publication in the journal in its current form, but we would like to consider a revised version that addresses the reviewers' and editors' comments. Obviously we cannot make any decision about publication until we have seen the revised manuscript and your response, and we plan to seek re-review by one or more of the reviewers. 

We expect to receive your revised manuscript by Jul 13 2020 11:59PM. Please email us (plosmedicine@plos.org) if you have any questions or concerns.

We look forward to receiving your revised manuscript. 

Sincerely,

Emma Veitch, PhD

PLOS Medicine

On behalf of Clare Stone, PhD, Acting Chief Editor,

PLOS Medicine

plosmedicine.org

*We'd suggest revising the title according to PLOS Medicine's style; this should have the initial phrase outlining the study question and then the study design(s) in the second phrase after a colon (eg, (: "randomized controlled trial," "A retrospective study," "A modelling study," etc.)

*In the last sentence of the Abstract Methods and Findings section, please describe the main limitation(s) of the study's methodology.

*At this stage, we ask that you include a short, non-technical Author Summary of your research to make findings accessible to a wide audience that includes both scientists and non-scientists. The Author Summary should immediately follow the Abstract in your revised manuscript. This text is subject to editorial change and should be distinct from the scientific abstract. Please see our author guidelines for more information: https://journals.plos.org/plosmedicine/s/revising-your-manuscript#loc-author-summary

*Please ensure the Methods section states whether the analysis plan followed in this paper for the 3 different studies was set out prospectively (ie prior to collection of data). Please state this (either way) early in the Methods section.

*Referencing callouts should ideally be sequential numerals in square brackets (ie [1], [2] etc). If referencing software was used then this should be fairly quick and easy.

*Please note the comments from one reviewer that the conclusions drawn in the paper should be more cautious, given the possible risks of multiple testing and the potential for false-positive findings (ie that some analyses may not survive correction for multiple testing).

Comments from the reviewers:

Reviewer #1: "Bile acid synthesis and modulation are associated with brain amyloid deposition, white matter lesions, neurodegeneration and risk of vascular dementia" employs a three-step approach to investigate possible relationships between primary bile acids (BA) and various biomarkers of Alzheimer's Disease (AD) and vascular dementia (VaD). Step 1 was concerned with serum and neuroimaging markers, Step 2 with exposure to bile acid sequestrants (BAS), and Step 3 with assessment of autopsied brain tissue.

The major experimental findings were that there possibly exists a sex-specific association between lower serum BA, and faster accumulation of brain white matter lesions (WML)/faster rate of brain atrophy in males (opposite effect in females; Step 1), supported by large-scale primary care data analysis (again for males; Step 2). A possible mechanism was then proposed with evidence that the principal BA receptors were significantly more often expressed in postmortem AD brains, compared to control brains (Step 3).

The authors have largely acknowledged the relative strengths and limitations of this study in the discussion, in particular the relatively small sample sizes involved in Step 1 and Step 3 (although the number of dementia cases [N=72] amongst BAS users is also relatively small, for Step 2). Overall, the manuscript represents a comprehensive and ambitious effort to quantify and describe potential mechanisms behind bile acid synthesis/modulation and AD/VaD signs. However, there remain some issues that might be addressed:

1. It is summarized in the abstract that "We found that lower serum concentrations of 7α-OHC, CA and CDCA were associated with higher brain amyloid deposition, faster WML accumulation and faster brain atrophy in males. Opposite effects were observed in females". However, the actual data presented in Table 1 suggests that this description might be a simplification that possibly mischaracterizes the specifics. For example, it appears that no statistically-significant correlations between 7α-OHC/CA/CDCA and amyloid deposition was found for females, as was the case for CA/CDCA and amyloid deposition for males. Moreover, the Results section then states that "We observed no significant associations in the female only-sample" (line 298). The specific findings might thus be presented more precisely in the abstract.

2. Perhaps of greater concern is that when analyzing all subjects (male+female) in Step 1, it appears that only 7α-OHC shows statistically-significant correlations with amyloid deposition/brain atrophy, with the actual BAs (CA/CDCA) exhibiting no correlations at all. The BA correlations appear only when considering a male/female sex-specific stratification of the cohort.

As such, it might be clarified whether this sex-specific analysis was a priori defined in the initial study design from some theoretical motivation/prior work, or if it was a post-hoc discovery. This is of interest due to the relatively small sample size (N=134/141) involved in Step 1, and also previous similar cited work on AD [citation 8] not automatically considering sex-based stratification.

3. For Step 2, the authors might consider discussing the appropriateness of using matched LMT users as the control group for BAS users, in greater detail. In particular, what might be some considerations leading to a patient being prescribed LMT instead of BAS (or vice versa)? This is particularly since the matching was performed based on relatively few demographics (sex, year of birth, region, year of clinic registration/first prescription)

4. For Step 2, a correlation was found between BAS use and VaD for males, but not between BAS use and any dementia/AD/other dementia, even for males. From line 120, the hypothesis inspired from Step 1 was whether BAS would alter risk of (various types of) dementia. As such, it might be discussed further as to whether there are any characteristics particular to VaD that might have given rise to this correlation, all the more due to the acknowledged limitation of likely inaccuracy in clinical diagnoses of dementia subtypes in primary care settings (line 481).

5. For Step 3, given the major role played by sex-based stratification in the previous analyses, it might be appropriate for the relevant analyses and results (Figures 5 & 6, Supplementary Tables 7 & 8) for this step to be stratified by age too.

6. It is stated that "these findings are among the first to demonstrate sex-specific associations between the rate-limiting step in primary BA synthesis and brain amyloid deposition as well as longitudinal changes in brain WML burden" (line 411). Given the significance of the sex-specific analyses throughout, it might be appropriate to discuss any other work suggesting possible sex-specific associations/mechanisms. The existing discussion does not appear to propose any plausible explanation for the observed sex-specific associations through the underlying mechanism.

Reviewer #2: In this manuscript, Varma VR. Et al aim to investigate the role of cholesterol catabolism in the pathogenesis of dementia. 

The study is divided in 3 steps:

-Step 1: the authors measured levels of primary bile acids (cholic acid CA, Chenodeoxycholic acid CDCA) and one of the main BAs precursor, 7alpha-hydroxycholesterol (7a-OHC) in the Baltimore Longitudinal Study of Aging (BLSA) and analyzed their association with amyloid deposition and neuroimaging data. Main findings: lower concentration of 7a-OHC, CA and CDCA are associated with increase of amyloid deposition, faster WML accumulation and brain atrophy in males. The opposite was observed in females.

-Step 2: the authors tested whether patients using bile acid sequestrants (BAS) had altered risk of dementia, by using the Clinical Practice Research Datalink (CPRD) dataset. Main findings: exposure to BAS increases risk of vascular dementia in males and not in females.

-Step 3: the authors investigated levels of BAs and 7OHC in autopsy brains from BLSA cohort and used gene expression omnibus (GEO) data to test whether mRNA levels of BAs receptors were altered in Alzheimer's Disease and control patients. Main findings: receptors for BAs, as FXR and TGR5, as well as concentration of CA and CDCA, are higher in AD brains.

The current study is relevant, and the research question is important in the field of metabolism of cholesterol as a risk factor for different types of dementia. The results have the potential to provide an advance over existing knowledge and also important implications for developing new therapeutic targets. The rationale behind is strong but data as presented may not be enough strong to support the hypothesis of the authors. There are critical points that should be addressed before the manuscript can be accepted for publication.

STEP 1. There are previous published results on this same topic supporting the hypothesis that circulating BAs may contribute to AD pathogenesis (PMID:19288586), where it is shown that low levels of CA are associated to greater atrophy and to reduced glucose metabolism (PMID:30337152) and that BAs serum levels are decrease in AD patients from ADNI cohort (PMID: 30337151). When compared to these previous studies, the authors of this manuscript added a number of novelties, such as using another study cohort, white matter lesions measurements, Pib +/- subjects, sex stratification. 

1) Were the samples collected from fasting patients? The information seems to be missing in the text.

2) As acknowledged by the authors the smaller N in this cohort may be a limitation and sex-specific analyses can further reduce power. It would be therefore very relevant to have data from other cohorts, ie ADNI, where some BAs were previously measured but sex stratification analyses were not provided (PMID: 30337152).

3) Were the analyses adjusted for use of medications for ie. cholesterol? This could also be very relevant and affect the analyses.

4) What are the levels of BAs in AD serum samples? Were the results from previous studies confirmed?

STEP 2

1) Users of statins were excluded, I wonder if it could be relevant to investigate them as well in this study? 

STEP 3

1) LOD of BAs in measurement from autopsy brain may be a limitation. Would it be possible to measure amyloid beta levels as well? It would add important information.

2) Use of data from GEO datasets to prove hypothesis. The authors are strongly suggested to perform qPCR on autopsy brain from BLSA cohort to investigate and confirm the changes in gene expression observed in datasets from GEO. They should use for this the same regions used for BAs measurements (brain areas used for gene expression of FXR and TGR5 are not the same, nor the same cohort).

3) Gene expression levels of receptors may not be enough to support hypothesis on mechanisms and at least protein levels should be also measured.

4) The results from STEP 3 do not support sex differences observed in Step 1 and 2, which are not explored. The authors should run analyses in males and females and if possible, increase the N for this study.

In general:

1) The differences in male and females in STEP 1 and 2 are the main novel result, very interesting and should definitely be discussed more and investigated deeper. What could be the mechanisms behind? Is there literature that could support these sex-specific differences? What reported by the authors seems not exhaustive.

Reviewer #3: Review comments on manuscript PMEDICINE-D-20-00371_R1

The manuscript by Varma et al explores the relationships between bile acid (BA) levels/synthesis and dementia related pathology, such as white matter lesions (WML) and amyloid deposition, as well as vascular dementia risk and sex related differences. To address these questions, the authors used three different experimental and/or analysis approaches: (1) a cross-sectional study to evaluate associations of serum bile acids with brain amyloid accumulation, WML and sex differences (BLSA cohort, n=141 - 134 subjects), (2) a subset from large clinical dataset (CPRD, UK) was used to test dementia risk in subjects using bile acid sequestrants (BAS) with n=3,208 BAS users and n=23,483 non-statin lipid modifying therapies (LMT) users, (3) cross-sectional study comparing concentrations of bile acids and mRNA expression levels of their receptors (FXR and TGR5) in control vs AD post-mortem brain samples (BLSA autopsy program, n=13 control and n=16 AD). 

The question of whether there is a relationship between peripheral levels of bile acids and dementia incidence or risk is an interesting one, particularly as bile acids are products of cholesterol metabolism, which has been widely studied regarding its role in neurodegenerative diseases of ageing. It is an emerging area of interest with relatively few studies exploring the role of bile acids in dementia, so the study asks a relatively novel question.

My concerns with the current work are as follows:

(1) On the back of three experiments of varying design and mostly all low powered studies, the authors are making some fairly strong claims about dementia associations and mechanisms. However the data to support these claims are low powered (as they themselves recognize and point out within their limitations section), and the p values frequently borderline, and unlikely to survive correction for multiple testing (e.g., Table 1 showing associations of BA with brain volumetric measures; cDVR, precuneus DVR, WML, Table 2 testing association of BAS use and dementia risk; out of 24 tests, only one has a p value of < 0.05, which would not survive correction for multiple testing, Fig 5 testing brain BA concentration in control vs AD subjects; out of 6 tests only one has p<0.05). These data are at best suggestive, and the discussion should reflect this, rather than presenting strong and definitive claims about dementia associations and mechanisms. Having a limitations section is a good idea, but does not obviate the need for a measured discussion, which makes claims in proportion to the strength of the data. 

(2) In several parts of the manuscript the authors claim to have used a "novel study design". However the three study designs they outline are commonly used study types. The authors can claim perhaps to have used 3 studies which each differ in design and/or methodology, but should avoid claiming/overstating the novelty of the design or methodology. Furthermore, I am uncertain as to whether combining three independent studies adds statistical power, or risks conflating potentially unrelated results? Using replication cohorts would have seemed a stronger approach, rather than using multiple small studies of orthogonal design. However a statistician would be better placed to address this question. 

(3) The authors interpret the lower BA values they observe in males as a possible "mediator of early pathological changes in AD" (pg 18). However they do not canvas other possible explanations, such as secondary effects, resultant from neuronal cell death, brain atrophy or neuropathologies present. In the case of the post mortem tissue, ease of metabolite extraction from control vs AD tissue might be another contributing factor (were internal standards used for the assay?). Furthermore the negative associations of metabolites and brain atrophy in females are not discussed in any detail, nor why the marked difference between males and females, nor reasons for the heterogeneous associations of brain pathology and specific metabolite levels. The rational for stratification by sex is also not entirely clear, since, of the three metabolites assayed, only 7�-hydroxycholesterol had a significant p-value of 0.034 (without correction for multiple testing) for the "total" group (Table 1). Similarly in Table 2/supplementary Table 4 there are no significant differences in the "overall" analysis, so the rationale for stratification should be more clearly outlined.

My specific comments are as follows:

- Subject numbers should be shown within all tables and figures, both in the main text and supplementary section, and include the totals and numbers following sex stratification. 

- Supplementary Table 2 shows that there are no significant associations between any of the three serum metabolite concentrations and PiB PET status, whereas Table 1 in the main text (pg 12) shows some significant associations between cDVR (amyloid-� deposition) and 7�-hydroxycholesterol in males. These apparently conflicting outcomes should be discussed, and some explanation offered. 

- It is a little odd that the authors discuss their findings in the context of "early neuroimaging markers", "early neurodegeneration", "early pathology changes", etc, since the majority of their subjects are >65 years of age, and well into the range of late onset dementia. 

- The authors do not make a clear statement as to whether the changes observed here are likely specific to a particular dementia subtype (they mention BA associations in both AD and vascular dementia), a particular neuropathology (amyloid deposition, WMH) or may just be an age related phenomenon, increasing in parallel with increase of neuropathological features with age.

- Table 2 and supplementary Table 4 appear to have the same data, but supplementary Table 4 has more detail. I suggest replacing Table 2 with supplementary Table 4 in the main manuscript.

- The Figure 3 flow chart shows that n=3,208 BAS users and 23,483 LMT users were available from the CPRD dataset. However were all these subjects used in the analyses presented here? It is a bit difficult to ascertain since subject numbers were not shown in most figures and tables (see previous comment - they should all be shown) - however the "N cases" in supplementary figure 4 have much smaller numbers. If fewer numbers were used for analysis than shown at the bottom of figure 3, then it would be useful to extend this flow chart to show the number of BAS and LMT users that were included in the analyses performed in the current study. 

- The authors are using a non-conservative approach to data/results interpretation, frequently claiming that specific associations were established, when only uncorrected data have p<0.05 but correction for multiple testing /FDR values not are statistically significant. Occasionally they even claim to observe changes when the uncorrected p-value is ≥0.05. This approach is particularly evident on pg 14 (results section). While they could speak of suggestive changes or data trends, however data which does not survive correction for multiple testing, or has p-values ≥0.05 should be reported as not significant. There is otherwise a risk of overinterpreting weak data. 

- Figure 6 would more clearly represent the level of difference between control and AD bile acid gene expression if the y-axes were presented as non-transformed data, rather than log2 data. 

- The last paragraph on pg19 beginning "Our study design represents…." could be deleted, since a detailed outline of the three studies was provided earlier in the manuscript.

[LINK]

---

## [Decision Letter · Decision Letter 2]

25 Nov 2020

Dear Dr. Varma,

Thank you very much for submitting your revised manuscript "Bile acid synthesis and modulation are associated with brain amyloid deposition, white matter lesions, neurodegeneration and risk of vascular dementia: a metabolic, neuroimaging, pharmacoepidemiologic and transcriptomic analysis" (PMEDICINE-D-20-00371R2) for consideration at PLOS Medicine. 

Your revision was evaluated by a senior editor and discussed among the editors here. It was also discussed with an academic editor with relevant expertise, and sent to two of the original reviewers, including the statistical reviewer. The reviews are appended at the bottom of this email and any accompanying reviewer attachments can be seen via the link below:

[LINK]

In light of these reviews, I am afraid that we still will not be able to accept the manuscript for publication in the journal in its current form, but we would like to consider a further revised version that addresses the reviewers' and editors' remaining comments. Obviously we cannot make any decision about publication until we have seen the revised manuscript and your response, and we plan to seek re-review. 

We expect to receive your revised manuscript by Dec 16 2020 11:59PM. Please email us (plosmedicine@plos.org) if you have any questions or concerns.

We look forward to receiving your revised manuscript. 

Sincerely,

Thomas McBride, PhD

Senior Editor 

PLOS Medicine

plosmedicine.org

1- Please report your study according to the relevant guideline(s), which can be found here: http://www.equator-network.org/ Please include the completed checklist as Supporting Information. When completing the checklist, please use section and paragraph numbers, rather than page numbers. Please add the following statement, or similar, to the Methods: "This study is reported as per the XXX guideline (S1 Checklist)." As this study uses multiple designs, I am leaving it to you to choose the appropriate guideline(s).

2- In your data statement, please include accession numbers or DOIs necessary for researchers to request the specific datasets used in this study.

3- Thank you for editing the reference call-outs. Please also remove the space between reference numbers (e.g., [1,2]).

4- Thank you for editing the title, however, please edit further to fit PLOS Medicine’s style. Titles cannot be declarative. Additionally, while we realize that this is a unique multi-step study design, but a briefer description of the study design would be preferred.

* Throughout the manuscript, please remove causal language (e.g., “increases risk of” at line 72 should be “was associated with”).

5- Please edit the Abstract Background section to replace the summary of the 3 step study design with a brief statement of the main study question. The study design should be introduced in the Abstract Methods and Findings section.

6- Abstract Methods and Findings:

* Please include the population and setting, years during which the study took place, length of follow up, and brief demographic details (e.g., age, sex) for each cohort included.

* Please quantify the main results (with 95% CIs and p values).

* Before describing the positive association at line 71, lease add a sentence describing the negative findings quoted around line 505.

* Please remove the last sentence (“Our findings merit confirmation…”) or move it to the Abstract Conclusions

7- The Abstract Conclusions are still a bit strong, given the limitations of this study. Please address the study implications without overreaching what can be concluded from the data; the phrase "In this study, we observed ..." may be useful.

* Please interpret the study based on the results presented in the abstract, emphasizing what is new without overstating your conclusions.

* Please remove the last sentence “Targeting brain primary BA signaling...”

8- Figure 2 can be removed, and Figure 1 can be a supplementary figure.

9- In the Methods section, please include a section on ethics approval, specify the ethics boards/committees that provided approval for each of the cohorts and studies included (rather than “The local Institutional Review Board...”, and specify how consent was obtained (or who waived consent, if that was the case).

10- Line 653, “(fig-xx)” is this a reference to a figure to be added?

11- Final sentence, lines 712-714, please be a bit more specific on the implications for clinical practice and drug discovery. 

12- Please move the funding information from the main text to the Financial Disclosure section of the metadata.

13- At the end of the main text, please provide a list of all supplementary documents with titles and legends where applicable. Please also make sure all supplementary documents are referenced when relevant to the main text.

Comments from the reviewers:

Reviewer #1: We thank the authors for largely addressing our concerns from the previous review round. A fairly major concern however remains with the sex-specific associations claimed with CA and CDCA. In particular, with reference to the additional confirmatory ADNI analysis as presented in Table 1b, the impression gained at first glance is that the associations with CA and CDCA are universal (from all variables being significant in the "Total (n=1666)" column). Indeed, looking at the pvals, it appears likely that the non-significance of many variables once broken down into male/female subsets, might have arisen largely due to the reduced sample size.

Furthermore, while there appear to be far more CA/CDCA variables showing significance in Table 1b for males as compared to females, it is notable that all these nine variables have pval (FDR)=0.049, which appears to be just marginally below the significance threshold of p=0.05. In other words, were the significance threshold changed by merely 0.002, the conclusion reached would have been that neither CA nor CDCA variables had been shown to have had any associations with the rate of brain atrophy for males. Given that p=0.05 is a somewhat arbitrary, if traditional, measure (see for example "The reign of the p-value is over: what alternative analyses could we employ to fill the power vacuum?", L.G. Halsey, Biology Letters, 2019), it might remain slightly problematic to claim evidence of general CA/CDCA sex-specific associations as a major finding on this basis. Nonetheless, we believe the rest of the analysis to have merit.

Minor issue: in Line 179, "Religions Orders" might be "Religious Orders".

Reviewer #2: Dear authors, 

Thank you for your detailed responses and additional data. It added considerable value to the manuscript.

I don't have any further comment.

Best regards

Reviewer

[LINK]

---

## [Decision Letter · Decision Letter 3]

23 Feb 2021

Dear Dr. Varma,

Thank you very much for re-submitting your manuscript "Bile acid synthesis, modulation and dementia: a metabolomic, transcriptomic and pharmacoepidemiologic study" (PMEDICINE-D-20-00371R3) for consideration at PLOS Medicine. We do apologize for the delay in sending you a response. 

I have discussed the paper with our academic editor and it was also seen again by one reviewer. I am pleased to tell you that, provided the remaining editorial and production issues are dealt with, we expect to be able to accept the paper for publication in the journal.

[LINK]

Please let me know if you have any questions, and we look forward to receiving the revised manuscript shortly.   

Sincerely,

Richard Turner PhD, for Thomas McBride, PhD

rturner@plos.org

Requests from Editors:

In your data statement, please use "data are" consistently. Are all sources of data for your study quoted here (ADNI appears to be absent)? It appears that information on ADNI needs to be moved from the Acknowledgements section at the end of the paper. 

At line 91, should "CN" be "CON"?

At line 133, please remove "Finally ..."; and we suggest adapting the wording to "... levels of gene expression of BA receptors were altered ...". 

In your Introduction around lines 160-175, please integrate "STEP 1" and so on into the text in a single paragraph; e.g., "First, we used targeted metabolomics assays ... Finally, we explored plausible molecular mechanisms ...".

At line 539, we ask you to adapt "marginally increased risk" to "non-significantly elevated risk" (removing the subsequent "(non-significant)", noting the p value of 0.06.

In the paragraph summarizing the study findings at the start of your Discussion section, please make that "were/was associated" at lines 601/603. 

Please remove the information on data access from the end of the main text. This information will appear in the article metadata, via entries in the submission form. 

In the reference list, please ensure that journal names are abbreviated consistently (e.g., "J Alzheimers Dis." for reference 2 and others). 

For reference 22 and any other preprints cited, please substitute "[preprint]" for the sentence about the absence of peer review. 

For reference 44, please make the journal name "PLoS ONE".

Please break the STROBE checklist out into a separate attached file, labelled "S1_STROBE_Checklist" or similar and referred to by this label in your Methods section. 

Comments from Academic Editor:

This study reports a nice multi-pronged approach to explore hypotheses from a variety of angles. 

The authors should improve their discussion of what the implications of sex-specific pathways might be, the magnitude of possible differences, and how this might fit with our understanding of descriptive epidemiological estimates of dementia. 

The authors might also want to consider making stronger statements about the fact that so much research doesn't report or address sex. 

I'd like the authors to acknowledge literature that suggests their ideas of circumscribed AD, VaD are out of kilter with what's observed in usual dementia in the oldest age groups, thus contributing to the perpetuation of identifiable clean subtypes (reasonable for younger old, not for older old). 

Comments from Reviewers:

*** Reviewer #1: 

We thank the authors for clarifying our main concern about the FDR pvals, and reflecting a slightly more nuanced treatment in the Abstract and Discussion.

The claim however in the response that "the raw p-values for those comparisons are generally far lower in the male-only samples than the female-only samples" might however then be supported by these raw p-values, perhaps in the Supplementary material. Moreover, it might be explained as to why these raw p-values were not initially used for analysis in this case since they supposedly demonstrate sex-specific differences that much more clearly, and why the minimum alpha method of calculating false discovery rate was used instead.

***

[LINK]

---

## [Editor Report · Decision Letter 4]

6 Apr 2021

Dear Dr Varma, 

On behalf of my colleagues and the Academic Editor, Dr Brayne, I am pleased to inform you that we have agreed to publish your manuscript "Bile acid synthesis, modulation and dementia: a metabolomic, transcriptomic and pharmacoepidemiologic study" (PMEDICINE-D-20-00371R4) in PLOS Medicine.

Prior to final acceptance, we suggest amending the title to "... synthesis and modulation and risk of dementia: A metabolomic ..."; and removing "the" at line 50. 

PRESS

Sincerely, 

Richard Turner, PhD 

rturner@plos.org